# A reversible mitochondrial complex I thiol switch mediates hypoxic avoidance behavior in *C. elegans*

John O. Onukwufor[1,2], M. Arsalan Farooqi[1], Anežka Vodičková[1], Shon A. Koren [1], Aksana Baldzizhar[1], Brandon J. Berry[1], Gisela Beutner [3], George A. Porter Jr [2,3], Vsevolod Belousov[4,5], Alan Grossfield [6] & Andrew P. Wojtovich [1,2✉]

*C. elegans* react to metabolic distress caused by mismatches in oxygen and energy status via distinct behavioral responses. At the molecular level, these responses are coordinated by under-characterized, redox-sensitive processes, thought to initiate in mitochondria. Complex I of the electron transport chain is a major site of reactive oxygen species (ROS) production and is canonically associated with oxidative damage following hypoxic exposure. Here, we use a combination of optogenetics and CRISPR/Cas9-mediated genome editing to exert spatiotemporal control over ROS production. We demonstrate a photo-locomotory remodeling of avoidance behavior by local ROS production due to the reversible oxidation of a single thiol on the complex I subunit NDUF-2.1. Reversible thiol oxidation at this site is necessary and sufficient for the behavioral response to hypoxia, does not respond to ROS produced at more distal sites, and protects against lethal hypoxic exposure. Molecular modeling suggests that oxidation at this thiol residue alters the ability for NDUF-2.1 to coordinate electron transfer to coenzyme Q by destabilizing the Q-binding pocket, causing decreased complex I activity. Overall, site-specific ROS production regulates behavioral responses and these findings provide a mechanistic target to suppress the detrimental effects of hypoxia.

[1] Department of Anesthesiology and Perioperative Medicine, University of Rochester Medical Center, Rochester, NY 14642, USA. [2] Department of Pharmacology and Physiology, University of Rochester Medical Center, Rochester, NY 14642, USA. [3] Department of Pediatrics-Division Cardiology, University of Rochester Medical Center, Rochester, NY 14642, USA. [4] Center for Precision Genome Editing and Genetic Technologies for Biomedicine, Pirogov Russian National Research Medical University, 117997 Moscow, Russia. [5] Federal Center of Brain Research and Neurotechnologies, FMBA, Moscow 117997, Russia. [6] Department of Biochemistry and Biophysics, University of Rochester Medical Center, Rochester, NY 14642, USA. ✉email: Andrew_Wojtovich@urmc.rochester.edu

Organisms sense and respond to environmental oxygen levels in order to thrive. Oxygen, as an essential component for respiration and survival, serves to fuel mitochondrial function and metabolism. Low oxygen tension, or hypoxia, poses a challenge to survival since energy availability and demand become imbalanced. For example, the rapid reintroduction of oxygen from hypoxic to normoxic conditions, causes oxidative stress and underlies many diseases such as stroke, kidney injury, and myocardial infarction[1,2]. One way that organisms adapt to low oxygen levels is through compensatory behaviors, such as the avoidance of hypoxic environments[3]. The nematode *Caenorhabditis elegans* surveils the environment and responds to changes in oxygen concentration or oxidants by increasing their locomotory rate to increase the probability of escaping the hypoxic stimulus, herein referred to as hypoxic avoidance behavior. Reactive oxygen species (ROS) have been suggested as mediators of this hypoxic avoidance behavior through stimulating or inhibiting neuronal responses[4,5], though it remains unclear how ROS can have these opposite effects in vivo.

The mitochondrial electron transport chain enzymes, such as complex I, are a major source of ROS. The site and quantity of ROS produced depend on the environmental conditions. Under pathological conditions, ROS production is exacerbated and induces cellular damage and death. In hypoxia-reoxygenation injury, acute mitochondrial dysfunction causes overproduction of complex I ROS upon reoxygenation and initiates cell death[6,7]. Conversely, ROS also act as adaptive pro-survival signaling molecules during stress and hypoxic conditions[7–10]. This adaptive signaling potential of mitochondrial complex I ROS production has been largely overshadowed by research focused only on its damaging roles, obscuring potential novel avenues to target pathologic metabolism.

Here, we report that mitochondrial complex I senses hypoxia and induces avoidance behaviors through the oxidation of a single cysteine residue in the NDUF-2.1 subunit. This reversible oxidation decreases complex I enzymatic activity, presumably by destabilizing the coenzyme Q-binding pocket. We fused an optogenetic ROS-generating protein to an endogenous complex I ROS production site in *C. elegans* in order to produce site-specific complex I ROS. This enabled us to use light exposure to precisely control local complex I ROS signaling in vivo independent of other metabolic factors. While overproduction of complex I ROS is widely associated with hypoxic pathology, we demonstrate that site-specific ROS production instead results in a selective, physiologic pro-survival response.

## Results

**Locomotion in response to hypoxia and complex I ROS.** Oxygen levels inform *C. elegans* behavior[11]. When exposed to hypoxia, worms rapidly increase locomotion and over the course of minutes return to normal speed, even while remaining under hypoxia. When returned to normoxia, worms again increase locomotion (Fig. 1a). Given the previously reported associations between complex I ROS and hypoxia-reoxygenation, we hypothesized that mitochondrial complex I may be involved in this behavior. To first test if ROS are involved in the behavioral response to hypoxia, we exposed *C. elegans* to the superoxide dismutase/catalase mimetic EUK-134[12] and subjected them to acute changes in oxygen. We found that the behavioral responses to changes in oxygen were abolished by EUK-134, suggesting a role for ROS (Fig. 1a). We subsequently tested if ROS originating from complex I result in behavioral changes using the toxins paraquat and rotenone. Paraquat is a redox cycler that generates superoxide mainly through a mechanism involving complex I[13], while rotenone is a complex I inhibitor that results in increased

ROS production at complex I[6]. We used paraquat and rotenone to increase complex I ROS in vivo and measured changes in locomotion, finding that both paraquat and rotenone increased locomotion (Fig. 1b, c, Supplementary Fig. 1A–D). We tested if toxin-induced ROS is additive to hypoxia-induced ROS in mediating the behavioral response. We found that paraquat treatment increased baseline locomotion and suppressed the behavioral response to hypoxia. Upon re-oxygenation, paraquat-treated worms gradually returned to the increased locomotion seen pre-hypoxia (Supplementary Fig. 1E). These results highlight the difference in acute vs sustained ROS production. We hypothesize that the higher baseline ROS levels in paraquat-treated worms could prevent the dynamic changes necessary to respond to hypoxia. Alternatively, the higher levels of ROS could elicit compensatory mechanisms that block the acute response to hypoxia-mediated ROS. While supporting our hypothesis that ROS production at complex I leads to rapid behavioral changes, the toxins' effects are not acutely activated or reversible[6]. Therefore, we generated a system to study complex I ROS in vivo, independent of other metabolic factors and the irreversible application of toxins.

CRISPR/Cas9 genome editing was used to fuse the optogenetic ROS-generating protein Supernova to an endogenous site of ROS production, *nuo-1* (NADH:ubiquinone oxidoreductase-1; mammalian ortholog, *NDUFV1*), encoding the flavin-containing complex I subunit (Fig. 2a). As expected, the fusion was targeted to mitochondria, localized to the matrix compartment (Fig. 2b–d, Supplementary Fig. 2) and is incorporated into complex I (Supplementary Fig. 3). As suggested by our previous work on the biophysical properties of Supernova and other optogenetic ROS-generating proteins in vitro[14], the *nuo-1::Supernova* fusion allowed precise manipulation of mitochondrial complex I ROS production in vivo through regimented application of light (540–600 nm, Fig. 2e, f). Using dihydroethidium (DHE) detection of superoxide, we first confirmed the ability of *nuo-1::Supernova* to produce ROS in isolated mitochondria. Compared to DHE alone and wild-type controls, *nuo-1::Supernova* produced significantly greater amounts of superoxide, measured by HPLC separation of the superoxide specific marker of 2-hydroxyethidium (2-OHE$^+$) (Fig. 2e). Some superoxide was also detected at high doses of light in the wild-type controls, which is consistent with the expected effect of light exposure on DHE and biologic tissue (Supplementary Fig. 4)[14–16]. Superoxide, when produced in the mitochondrial matrix, can be converted to hydrogen peroxide ($H_2O_2$) in vivo[6]. We then targeted HyPer7, a $H_2O_2$ biosensor[17], to the mitochondrial matrix of both wild-type and *nuo-1::Supernova* worms to measure the amount of $H_2O_2$ produced upon illumination (Fig. 2f, Supplementary Fig. 4). *nuo-1::Supernova* generated more $H_2O_2$ than wild-type worms confirming our earlier results with isolated mitochondria.

After confirming that light induces ROS generation of *nuo-1::Supernova*, we returned to the behavioral model of complex I ROS signaling to dissect the molecular pathway. To test the functional impact of *nuo-1::Supernova*, we photoactivated Supernova and similarly assessed *C. elegans* behavior (Fig. 3). We found that light-induced complex I ROS decreased phototaxis (Fig. 3a, b) and increased locomotion (Fig. 3c, d), recapitulating the response from rotenone and paraquat treatment. This further supported the model of ROS production at complex I as a signal for increased locomotion and avoidance behavior. In order to fully characterize the physiological response to complex I ROS, we quantified several aspects of the photolocomotion response. Photoactivated *nuo-1::Supernova* animals had increased forward movement upon light exposure (Fig. 3e). Changes in direction were also significantly increased in photoactivated *nuo-1::Supernova* animals (Fig. 3f) while backwards movement was

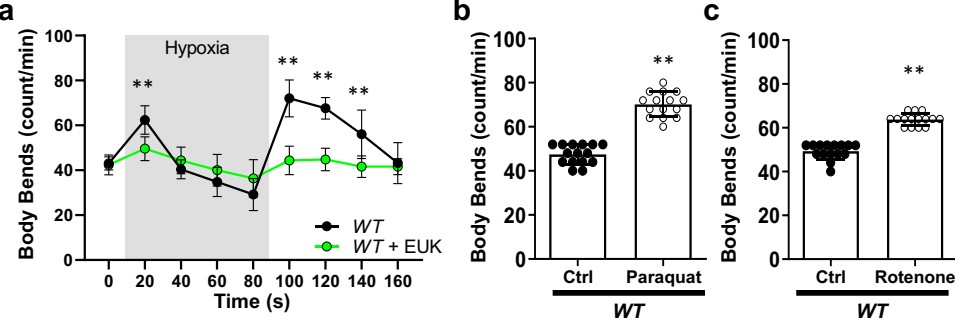

**Fig. 1 Hypoxia-mediated behavioral changes require reactive oxygen species. a** *C. elegans* increase locomotion in response to changes in oxygen concentration. Staged wild-type (N2 Bristol) L4 body bends were scored on unseeded plates in response to changes in oxygen. Baseline was recorded, and worms were then subjected to hypoxia and reoxygenation. Where indicated, worms were grown on plates containing the superoxide dismutase and catalase mimetic EUK134 (100 μM) 24 h prior. Data are mean ± SD. $N = 15$ independent animals across 3 technical replicates, $**p < 0.01$ (Two-way ANOVA, Sidak's multiple comparisons). **b, c** Complex I ROS increased *C. elegans* locomotion. Wild-type L4 worms were treated with (**b**) paraquat (1 mM) or (**c**) rotenone (1 μM) 24 h prior. Body bends were scored on unseeded plates. Data are mean ± SD. $N = 15$ independent animals across three technical replicates, $**p < 0.01$ (One-way ANOVA, Dunnett's multiple comparisons). Full dose–response in Supplementary Fig. 1.

unchanged (Fig. 3g). The increase in locomotion was light-dose dependent, plastic, and rapidly reversible (Fig. 3d, h). Further, continuous illumination sustained the avoidance behavior, and termination of illumination resulted in a rapid reversal back to normal locomotion (Fig. 3h). Importantly, the response to light exposure did not diminish with repeated trials, suggesting a high degree of biological turnover (Fig. 3h). Like many signaling pathways, the response diminished with age (Fig. 3i).

Additionally, we tested if the behavioral response was specific to complex I ROS production or rather due to mitochondrial ROS in general. We used CRISPR/Cas9 generated *complex II*[18] *and complex III Supernova* fusions to determine if light-induced ROS produced at other electron transport chain complexes could recapitulate the response (Supplementary Fig. 5). We subjected the light-induced electron transport chain ROS fusions to the phototaxis and behavioral assays and tested for a light-dose-dependent effect and negative phototaxis effect. While there were varying degrees of responses from all the strains, the magnitude of the *nuo-1::Supernova* response was not recapitulated by *complex II::Supernova* or *complex III::Supernova*, supporting the selectivity of local complex I ROS signaling (Supplementary Fig. 5). We also tested classical complex I (*gas-1(fc21)*) and complex II (*mev-1(kn1)*) mitochondrial mutants, which are characterized by increased oxidative stress. The complex II mutant responded to changes in oxygen concentration (Supplementary Fig. 6), suggesting, like the *complex II::Supernova* fusions, that complex II ROS does not affect hypoxic behavioral responses. The complex I *gas-1(fc21)* mutant, in contrast, had overall suppressed movement and no observable response to hypoxia. Overall, these results confirmed that many aspects of movement were affected by complex I ROS production, and support a model where acute, transient redox changes in the complex I microdomain act as a signal for physiologic avoidance.

**Complex I ROS production impacts bioenergetic functions.** Since ROS can alter mitochondrial function, we sought to characterize the bioenergetic effect of ROS production at complex I using *nuo-1::Supernova*. The loss of *nuo-1* is lethal in worms[19]. There was no indication that expression of the fusion protein had detrimental consequences on mitochondrial function, as brood size was normal and there were no overt phenotypic changes (Supplementary Fig. 7). Through respiratory analysis and mitochondrial enzyme functional assessment, we found no significant effect on baseline mitochondrial function in mitochondria isolated in vitro (Fig. 4). Next, we tested the effects on respiration in

response to light-generated complex I ROS. We measured oxygen consumption rates of both maximally fueled respiration (state 3) and substrate-depleted respiration (state 4) in order to assess respiratory control ratio (RCR). The RCR demonstrates the ability of mitochondria to respond to an energy demand above the basal leak conditions[20]. We performed these assays fueling respiration both through complex I substrates and complex II substrates to assess the selectivity of the *nuo-1::Supernova* effect to the complex I microdomain. Mitochondria were illuminated and then activity was assessed. In all cases, wild-type controls were not affected by the maximum light dose (Fig. 4). Under conditions of complex I respiration, the *nuo-1::Supernova* worms showed decreased state 3 respiration with increasing light dose, resulting in a light-dose dependent decline in RCR (Fig. 4a). This shows that increased ROS production from complex I decreases the ability of mitochondria to respire maximally under substrate-rich conditions. Complex II respiration and RCR values were not affected by photoactivation of *nuo-1::SuperNova* (Fig. 4b), demonstrating the spatially restricted function of our system, consistent with our behavioral results.

We then assayed enzymatic activity for complexes I and II to follow up on the respiratory results (Fig. 4c). As expected, complex II activity was not affected (Fig. 4d) while we observed light-dose dependent inactivation of complex I activity (Fig. 4e). Interestingly, this inhibition appeared stable after light removal in isolated mitochondrial membranes and contrasts the reversible in vivo behavioral phenotype. We hypothesize that isolated membranes lack an enzyme or cofactor necessary for reversal. Since the total activity of complex I was impaired, we further tested if the site of modification was proximal to the Q-binding site or the NADH-binding site. We found that the NADH-binding site activity was not affected by photoactivation of *nuo-1::Supernova* (Fig. 4f and Supplementary Fig. 8), suggesting that the ROS-sensitive site was downstream of the flavin-containing subunit. Given that complex I is the site of Supernova fusion and the ROS effect is selective to complex I, we hypothesized that complex I is therefore sensitive to the ROS generation. The loss of complex I activity could be attributed to degradation or disassembly of complex I or through a reversible inhibitory oxidative modification. To test these scenarios, we first resolved photoactivated mitochondria using clear native electrophoresis and found no change in complex I levels, suggesting light-mediated complex I inhibition is not the result of degradation (Supplementary Fig. 9). To test if the light-mediated complex I inhibition is reversible, we photoactivated *nuo-1::Supernova* and

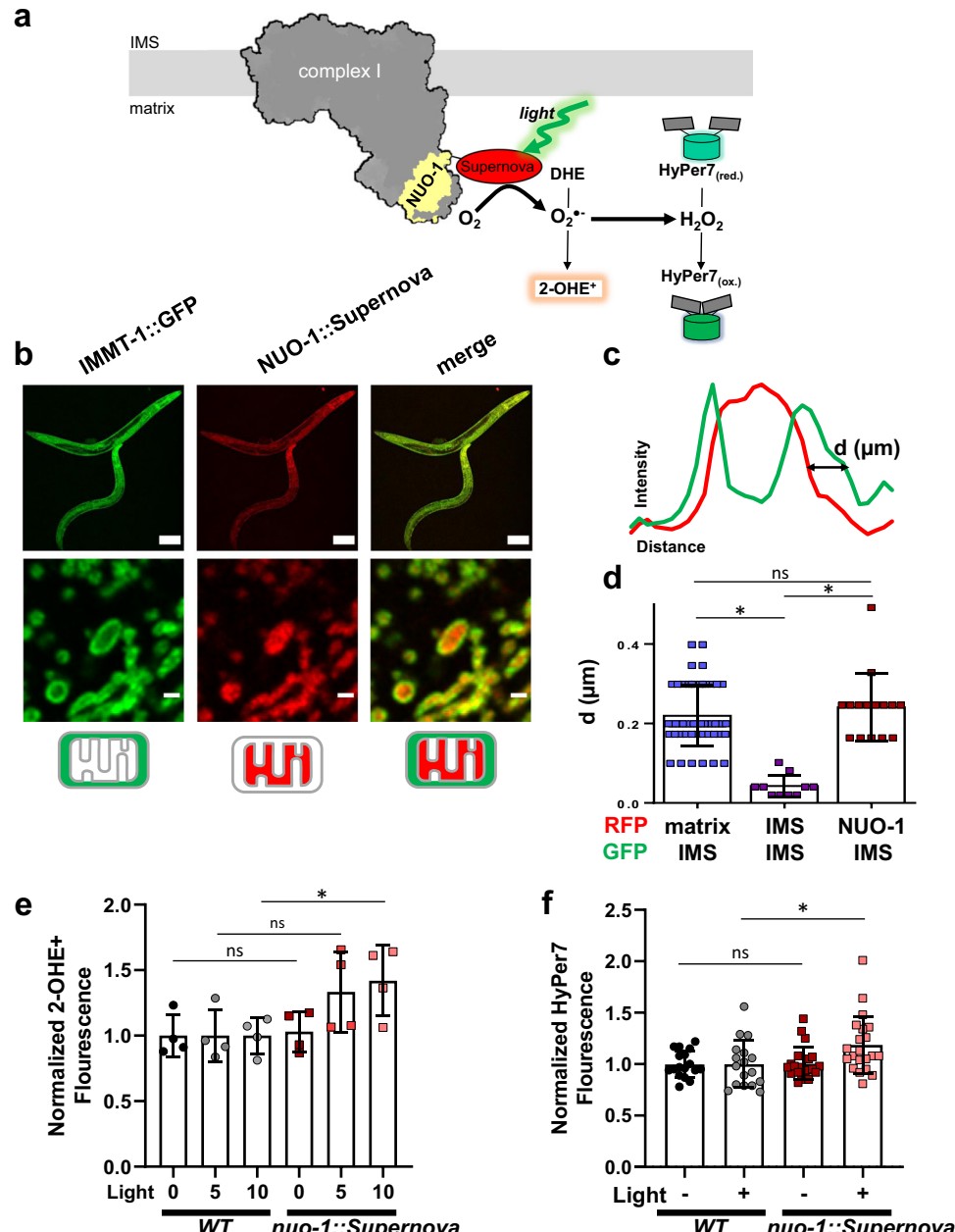

**Fig. 2 Optogenetic generation of complex I ROS production. a** Schematic illustration of light-induced complex I ROS generation. CRISPR/Cas9 fused Supernova to flavin mononucleotide (FMN) subunit, *nuo-1*. Upon illumination Supernova generates superoxide ($O_2^{\cdot-}$) which can be detected by the dihydroethidium (DHE) oxidation product 2-hydroxyethidium ($2\text{-}OHE^+$). Superoxide can then dismutate to hydrogen peroxide ($H_2O_2$) and is detected by the matrix-targeted biosensor HyPer7. **b** Localization of NUO-1::Supernova fusion protein. Mitochondrial electron transport chain complexes were tagged with fluorescent proteins. Compartmentalization of the fluorescent protein was assessed using the cristae maintenance protein IMMT tagged with GFP, which is restricted to outer membrane and inner membrane contact sites and not within cristae. Scale bar 100 (top) and 1 μm (bottom). Image is representative of at least three independent experiments. Additional localization data presented in Supplementary Figures. **c** Line scans of fluorescent protein signals across a mitochondrion. **d** Quantification of mitochondrial fluorescent protein fusion line scans. SDHB-1::mCherry, SDHC-1::mCherry, and IMMT::GFP are used as standards to characterize NUO-1::Supernova localization and are localized to the matrix, intermembrane space (IMS) and IMS, respectively. Individual line scan NUO-1::Supernova, SDHC-1::mCherry, and SDHB-1::mCherry provided in the Supplementary Figures. Data are mean ± SD. N = 50, 10, 15 independent mitochondria *p < 0.05 (Kruskal-Wallis test, Dunn's multiple comparisons). **e** Superoxide detection using $2\text{-}OHE^+$. Mitochondria isolated from wild-type and *nuo-1::Supernova* worms were illuminated (GYX, 7.8 mW/mm²) for the indicated time, in the presence of dihydroethidium (DHE, 100 μM) for $2\text{-}OHE^+$ separation using HPLC. Data normalized to WT for each condition. Raw data are in Supplementary Fig. 4A, B. Data are mean ± SD. N = 4 independent mitochondrial preparations, *p < 0.05, **p < 0.01 (Two-way ANOVA, Sidak's multiple comparisons). **f** Measurement of $H_2O_2$ with mitochondrial matrix-targeted HyPer7. L4 wild-type and *nuo-1::Supernova* worms were screened for bright pharyngeal expression of HyPer7. Worms were placed into glass-bottom 96-well plate containing M9 buffer and illuminated (GYX, 1.44 mW/mm²) for 2 min. HyPer7 intensity ratio (500/400 nm) are normalized to WT for each condition. Raw data are in Supplementary Fig. 4C, D. Data are mean ± SD. N = 17, 17, 21, 21 independent animals across 3 technical replicates, *p < 0.05, **p < 0.01 (Two-way ANOVA, Sidak's multiple comparisons).

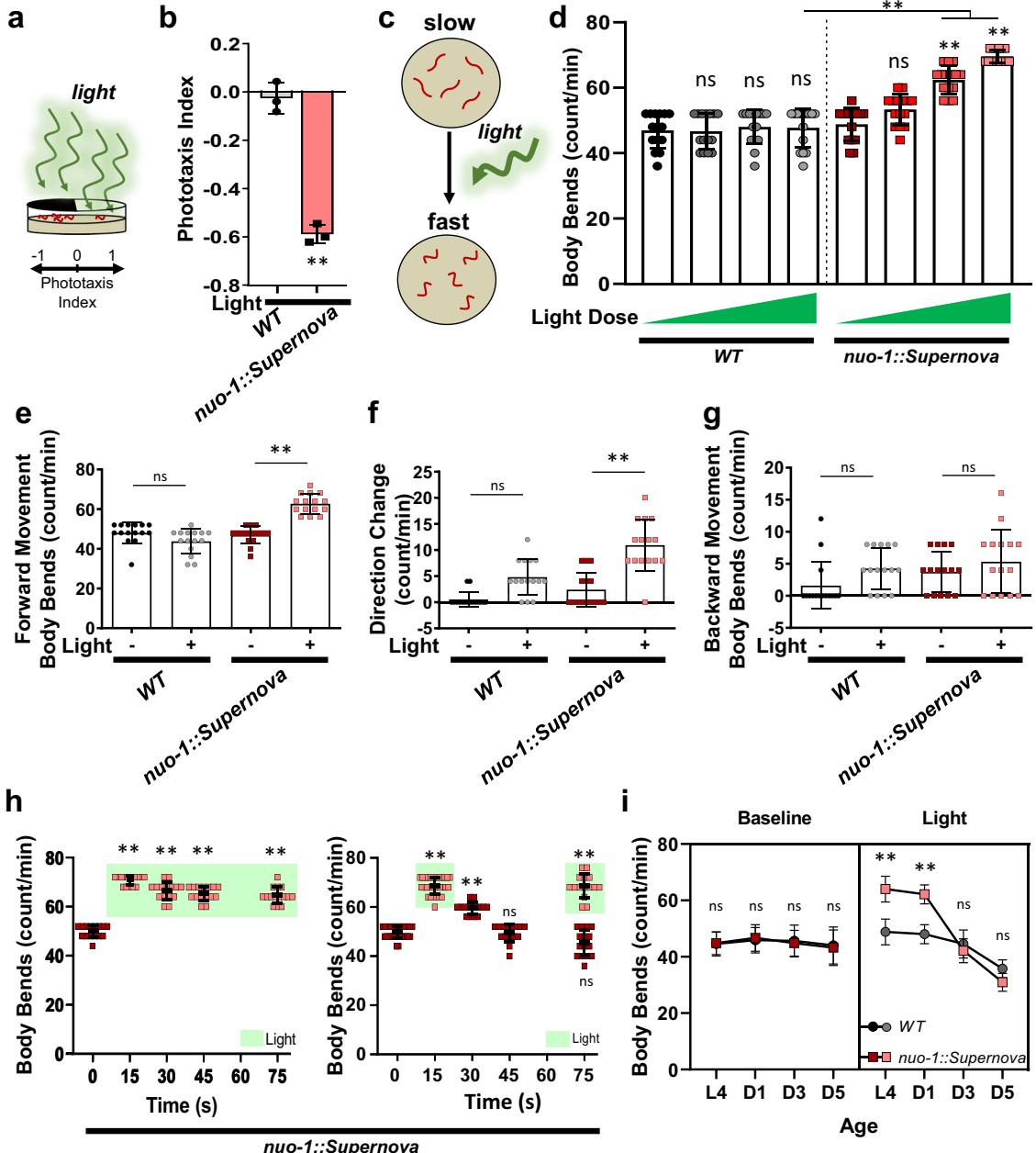

**Fig. 3 Characterization of Complex I ROS-induced avoidance behavior in *C. elegans*. a, b** Light-induced Complex I ROS increases phototaxis avoidance behavior *C. elegans*. **a** Schematic illustration of phototaxis experimental procedure. Wild-type (N2) and *nuo-1::Supernova* worms were transferred to the center of the seeded plate with half of the plate shaded (dark). Worms were then illuminated (GYX, 0.78 mW/mm$^2$) for 120 min and (**b**) the number of worms in the light and dark sections were scored and the phototaxis index was calculated. Data are mean ± SD. $N = 3$ independent experiments, each containing 25–80 animals, **$p < 0.01$ (unpaired, two-tailed *t*-test). **c–g** Light-induced Complex I ROS increases locomotion in *C. elegans*. (**c**) Schematic illustration of experimental procedure. *C. elegans* were acclimated to an unseeded plate and body bends were counted pre- and post-illumination. Wild-type (N2) and *nuo-1::Supernova* worms were individually transferred to unseeded plates and body bends were scored with and without light (MVX, 5.6 mW/mm$^2$) for 15 s on (**d**) light titration of whole locomotion (MVX, 0, 0.9, 2.5, 5.6 mW/mm$^2$); (**e**) forward movement; (**f**) Omega turn and (**g**) backward movement. Data are mean ± SD. $N = 15$ independent animals across 3 technical replicates, **$p < 0.01$ vs no light (Two-way ANOVA, Sidak's multiple comparisons). **h** Reversibility and plasticity of light-induced complex I ROS. *nuo-1::Supernova* worms were transferred to unseeded plates and body bends were scored with light (MVX, 5.6 mW/mm$^2$) for 15 s. For steady light, the light source was maintained and body bends were scored every 15 s for 60 s. For the reversal assay, the light source was removed and body bends were scored every 15 s for 60 s. For plasticity, the light was removed for 60 s, then reintroduced and body bends were scored for 15 s. Data are mean ± SD. $N = 15$ independent animals across 3 technical replicates, **$p < 0.01$ (Two-way ANOVA, Sidak's multiple comparisons). **i** Aging abolished complex I ROS induced increase in locomotion. Staged Wild-type (N2) and *nuo-1::Supernova* worms L4, Day 1, Day 3 and Day 5 were individually transferred to unseeded plates and body bends scored with and without light (MVX, 5.6 mW/mm$^2$) for 15 s. Data are mean ± SD. $N = 15$ independent animals across 3 technical replicates, **$p < 0.01$ vs wild-type (Three-way ANOVA, Tukey's multiple comparisons).

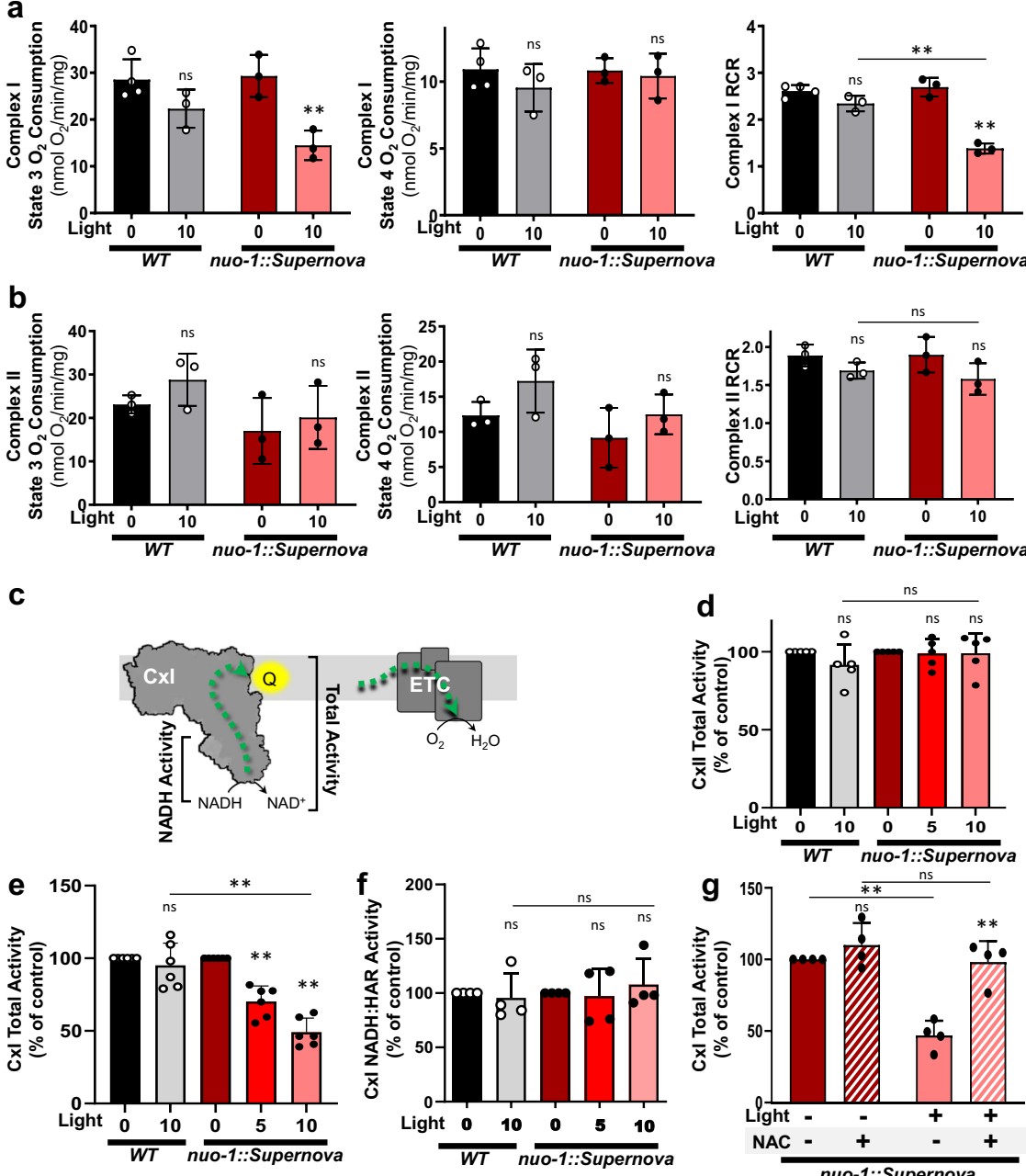

**Fig. 4 Light-induced complex I ROS effects on mitochondrial bioenergetics. a** Effects of light-induced complex I ROS on (**a**) complex I and (**b**) complex II respiration. Wild-type and *nuo-1::Superno*va isolated mitochondria were exposed to light (GYX, 7.8 mW/mm²) for 0 or 10 min. State 3, State 4 respiration, and the RCR were measured using (**a**) complex I- and (**b**) complex II-linked substrates. Data are mean ± SD. $N = 3$ independent mitochondrial preparations, ns = not significant, $*p < 0.05$, $**p < 0.01$ (Two-way ANOVA, Tukey's multiple comparisons). **c** Schematic diagram of the mitochondrial electron transport chain. Complex I transfers electrons to coenzyme Q which will ultimately reduce oxygen to water. Complex I activity was assessed via oxygen consumption, total activity (NADH to Q) and NADH dehydrogenase activity (NADH to hexaammineruthenium). **d** Effects of light-induced complex I ROS on complex II enzyme activity. Isolated freeze-thawed mitochondria were exposed to light (7.8 mW/mm²) for 0, 5, and 10 min. Complex II activity was measured as malonate-sensitive succinate oxidation rate of DCPIP reduction. Data are mean ± SD. $N = 5$ independent mitochondrial preparations, ns=not significant (Two-way ANOVA, Tukey's multiple comparisons). **e** Effects of light-induced complex I ROS on complex I total activity. Isolated freeze-thawed mitochondria were exposed to light (7.8 mW/mm²) for 0, 5, and 10 min. Total complex I activity was measured as the rotenone-sensitive rate of NADH oxidation. Data are mean ± SD. $N = 6$ independent mitochondrial preparations, $**p < 0.01$ (Two-way ANOVA, Tukey's multiple comparisons). **f** Effects of light-induced complex I ROS on complex I NADH dehydrogenase activity. Isolated freeze-thawed mitochondria were exposed to light (7.8 mW/mm²) for 0, 5, and 10 min. NADH dehydrogenase activity of complex I was measured as the hexaammineruthenium (HAR)-dependent rate of NADH oxidation. Data are mean ± SD. $N = 4$ independent mitochondrial preparations, ns = not significant (Two-way ANOVA, Tukey's multiple comparisons). **g** NAC reverses effects of light-induced complex I ROS on complex I total activity. Isolated freeze-thawed mitochondria were exposed to light (7.8 mW/mm²) for 0, 5, and 10 min. Total complex I activity was measured in the presence of 2.5 mM NAC. Note: NAC was not present during light treatment. Data are mean ± SD. $N = 4$ independent mitochondrial preparations, $**p < 0.01$ (Two-way ANOVA, Tukey's multiple comparisons).

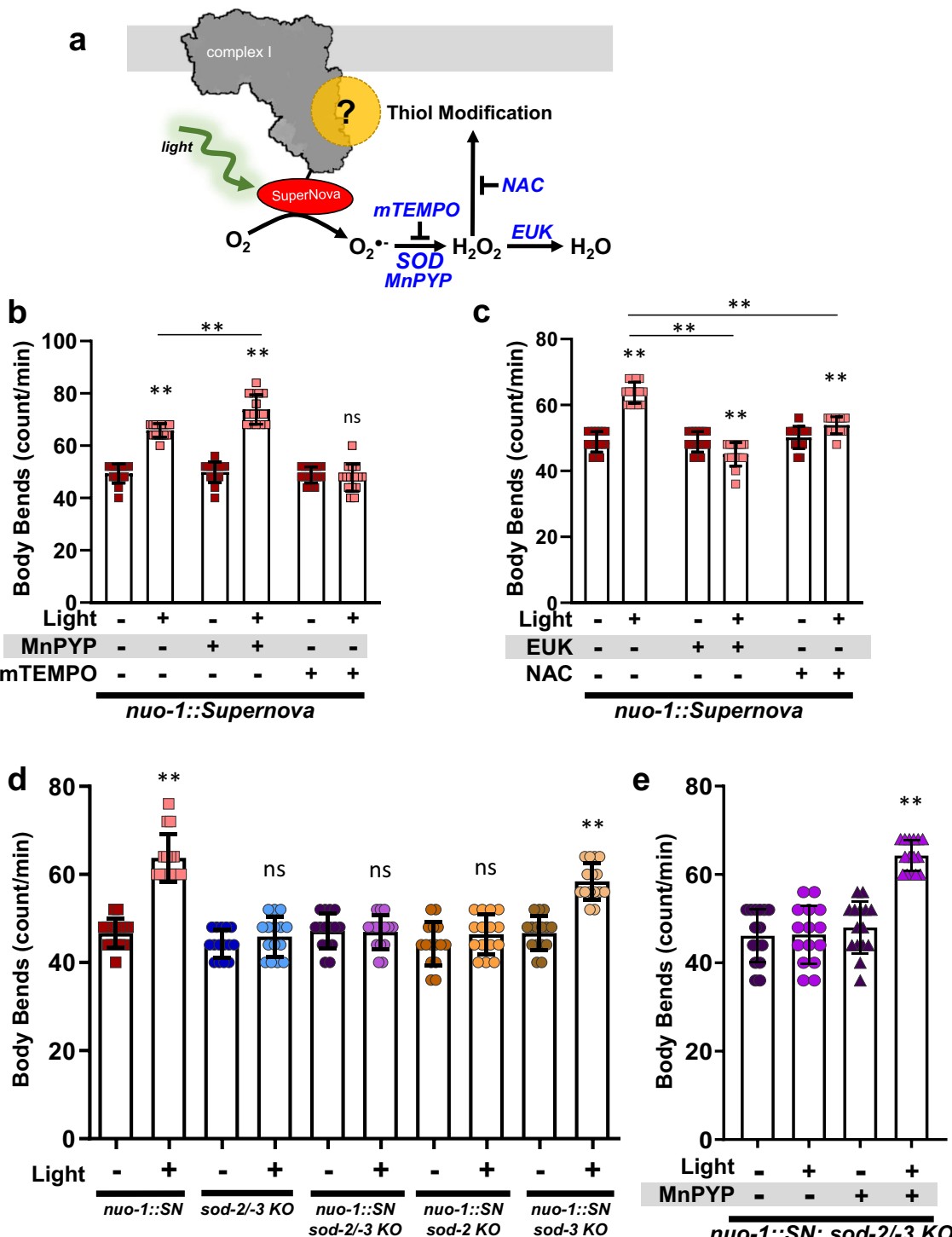

then measured complex I activity in the presence of a broad antioxidant, *N*-acetylcysteine (NAC). The light-induced inhibition of complex I activity was indeed reversible by NAC (Fig. 4g). Collectively, these results suggest a reversible redox modification in complex I downstream of the NADH-binding site and are consistent with time course of complex I protein turnover[21] and the rapid reversibility of behavioral responses (Fig. 3).

**Complex I ROS-mediated increase in locomotion.** To confirm the redox-dependence of these phenotypes, we used antioxidant and pharmacologic approaches in parallel to reverse the behavior (Fig. 5a, Supplementary Fig. 10). Supernova generates superoxide,

and the superoxide scavenger mitoTEMPO suppressed photo-avoidance in the *nuo-1::Supernova* strain (Fig. 5b) with no effect in wild-type controls (Supplementary Fig. 10). This demonstrated a requirement for superoxide generation. Superoxide dismutases are a class of antioxidant enzymes responsible for converting superoxide to hydrogen peroxide and *C. elegans* have two isoforms in the mitochondrional matrix, *sod-2* and *sod-3*. Using genetic knockout strains for mitochondrial SODs with *nuo-1::Supernova*, we tested whether the conversion of superoxide to hydrogen peroxide mediated ROS-induced locomotive responses. The loss of *sod-2* abolished the activation of avoidance through *nuo-1::SuperNova* photoactivation (Fig. 5d). Likewise, the addition of a

**Fig. 5 Pharmacologic and genetic modulations of complex I ROS induced changes in *C. elegans* locomotion. a** Schematic illustration of complex I ROS generation and sites of targeted modulations. **b** Effects of MnPyP (superoxide dismutase mimetic) and mitoTempo (mTEMPO) on complex I ROS-induced changes in *C. elegans* locomotion. *nuo-1::Supernova* were incubated with MnPyP (100 µM) or mTEMPO (10 µM) for 24 h. Worms were then individually singled and transferred into an unseeded plate and body bends were scored for 15 s with and without illumination (MVX, 5.6 mW/mm$^2$). Data are mean ± SD. *N* = 15 independent animals across 3 technical replicates, **p < 0.01 (Three-way ANOVA, Tukey's multiple comparisons). Full dose response and wild-type controls available in Supplementary Fig. 10. **c** Effects of EUK-134 (catalase mimetic, EUK) and NAC on complex I ROS-induced changes in *C. elegans* locomotion. *nuo-1::Supernova* were incubated with EUK (100 µM and NAC (2.5 mM) for 24 h. Worms were then individually singled and transferred into an unseeded plate and body bends were scored for 15 s with and without illumination (MVX, 5.6 mW/mm$^2$). Data are mean ± SD. *N* = 15 independent animals across 3 technical replicates, **p < 0.01 (Three-way ANOVA, Tukey's multiple comparisons). Full dose response and wild-type controls available in Supplementary Fig. 10. **d** Complex I ROS-induced changes in *C. elegans* locomotion requires *sod-2/-3*. L4 *nuo-1::Supernova*, *sod-2/-3*, *nuo-1::Supernova + sod-2/3*, *nuo-1::Supernova + sod-2*, and *nuo-1::Supernova + sod-3* worms were individually singled and transferred to an unseeded plate. Body bends were then scored for 15 s with and without illumination (MVX, 5.6 mW/mm$^2$). Data are mean ± SD. N = 15 independent animals across 3 technical replicates, **p < 0.01 (Two-way ANOVA, Sidak's multiple comparisons). **e** Superoxide dismutase mimetic rescues photolocomotion in the absence of *sod-2/-3*. *nuo-1::Supernova + sod-2/3* worms were incubated with MnPyP (100 µM) for 24 h. Worms were then individually singled and transferred into an unseeded plate and body bends were scored for 15 s with and without illumination (MVX, 5.6 mW/mm$^2$). Data are mean ± SD. *N* = 15 independent animals across 3 technical replicates, **p < 0.01 (two-way ANOVA, Tukey's multiple comparisons).

---

SOD mimetic, Mn(III)PrPYP, exacerbated light-induced ROS effects (Fig. 5b) and rescued the photolocomotion response in the absence of SOD (Fig. 5e).

Our results suggest that the formation of hydrogen peroxide is required for the behavioral response. Using a superoxide dismutase/catalase mimetic EUK-134 and NAC, we obtained similar results as hydrogen peroxide was required for the activation of avoidance upon *nuo-1::Supernova* activation (Fig. 5c). The effect of EUK-134 was not recapitulated with EUK-8, a structurally similar compound with less antioxidant activity (Supplementary Fig. 10D)[22]. Collectively, these experiments suggest that production of hydrogen peroxide in the region of complex I may be required for the physiologic avoidance signaling, and is consistent with redox signaling pathways.

**Hydrogen peroxide thiol modification through cysteine oxidation.** Our data supported a direct role of complex I ROS in mediating hypoxic signaling events; however, like many ROS-mediated biological processes, the protein targets that drive the hypoxic or behavioral phenotypes in vivo are unknown. The role of complex I ROS in hypoxia is widely studied in mammalian systems and we focused on the NDUFS2 subunit for structural analysis. The NDUFS2 subunit forms part of the mammalian coenzyme Q-binding site in complex I and is located in the mitochondrial matrix downstream of the NADH-binding site (Fig. 6a, b)[23]. The NDUFS2 subunit is required for respiratory supercomplex assembly[24], acute oxygen sensing[25,26], hypoxic signaling[27], and oxidative stress[24,28]. Many ROS modifications occur through reversible modification of protein cysteine (Cys) residues, though not all Cys residues are susceptible or biologically relevant to ROS-modification. All three NDUFS2 redox-active Cys residues defined in the recently published Cys redox proteome[29] are conserved in the *C. elegans* ortholog *nduf-2.1* (a.k.a. *gas-1* in *C. elegans*). In particular, mammalian Cys347 (Cys366 in *C. elegans*) is reversibly oxidized in models of hypoxia-reoxygenation and glutathione depletion[24,28] and is well-conserved (Fig. 6c, Supplementary Fig. 11A, B), though the role of this Cys oxidation is unclear. We sought to directly test the effect of *C. elegans* Cys366 oxidation in mediating behavioral responses to ROS.

Using CRISPR/Cas9, we mutated the redox-sensitive Cys366 to serine (Ser), a non-oxidizable Cys mimetic[30] and found no overt phenotypes (Fig. 6, Supplementary Fig. 7). When we activated *nuo-1::Supernova* to generate ROS, we found that there was no behavioral avoidance in the Cys366Ser mutant background (Fig. 6d). Moreover, rotenone-mediated ROS generation failed to increase locomotion (Fig. 6e, Supplementary Fig. 1d) and elicit

a phototaxis response (Fig. 6f) in the Cys366Ser strain, suggesting that Cys366 is necessary for the behavioral response to ROS. When we mutated redox-active Cys366 to aspartate to mimic a sulfinic acid modification, we found that this point mutation reduced brood size (Supplementary Fig. 7) and decreased complex I activity (Fig. 6g), mimicking a canonical *nudf-2.1* mutant[31]. Moreover, the Cys366Asp had increased baseline locomotion (Fig. 6d) similar to the toxin-ROS-activated phenotype (Fig. 1, Supplementary Fig. 1e). We next tested if the light-induced inhibition of complex I required Cys366. Using isolated mitochondria, we established that light failed to inhibit complex I activity in the Cys366Ser mutant background (Fig. 6h), supporting a role for Cys366 mediating redox changes in complex I activity and animal behavior. While our results do not preclude a role for other complex I Cys residues, they demonstrate that Cys366 was both necessary and sufficient to mediate ROS-induced behavioral changes.

We have experimentally established the role of Cys366, but the crystal structure of the mammalian protein (PDB: 6ZR2)[32] does not suggest an obvious mechanism. NDUFS2 coordinates both the Q-binding pocket and an iron-sulfur complex, N2, though these reside ≥25 Å away from the mammalian ortholog of Cys366, Cys347. Since Cys347 faces internally toward adjacent helices in NDUFS2 on α-helix 13 (Supplementary Fig. 11B), we used elastic network models to identify the dynamics of this protein in the context of the larger complex. This analysis revealed mammalian NDUFS2 contains a rigid bundle of four a-helices (α5-6-7-13), which partly encapsulate the Q-binding pocket and iron-sulfur-stabilizing region (Supplementary Fig. 11C, D). We speculate that oxidation at internally-facing Cys347/Cys366 on NDUFS2 could sterically disrupt this rigid helical bundle, destabilizing Q binding and thus decrease complex I activity as measured.

**A single thiol modification mediates hypoxic behavior.** Complex I ROS is canonically associated with hypoxia reoxygenation injury through the pathologic overproduction of ROS at reoxygenation[33]. However, in addition to the site of ROS generation, the timing and duration of ROS can result in distinct outcomes. For example, complex I ROS is also associated with oxygen sensing in mammals[3,25,26]. Given the hypoxic signaling role our results demonstrate, we hypothesized that complex I ROS can initiate signaling that acutely alters behavior to protect against hypoxic stress. We first tested if the *nduf-2.1* mutants could respond to acute changes in oxygen (Fig. 7a). The Cys366Ser mutation mimicked antioxidant treatment and failed to respond to changes in oxygen levels. Interestingly, upon

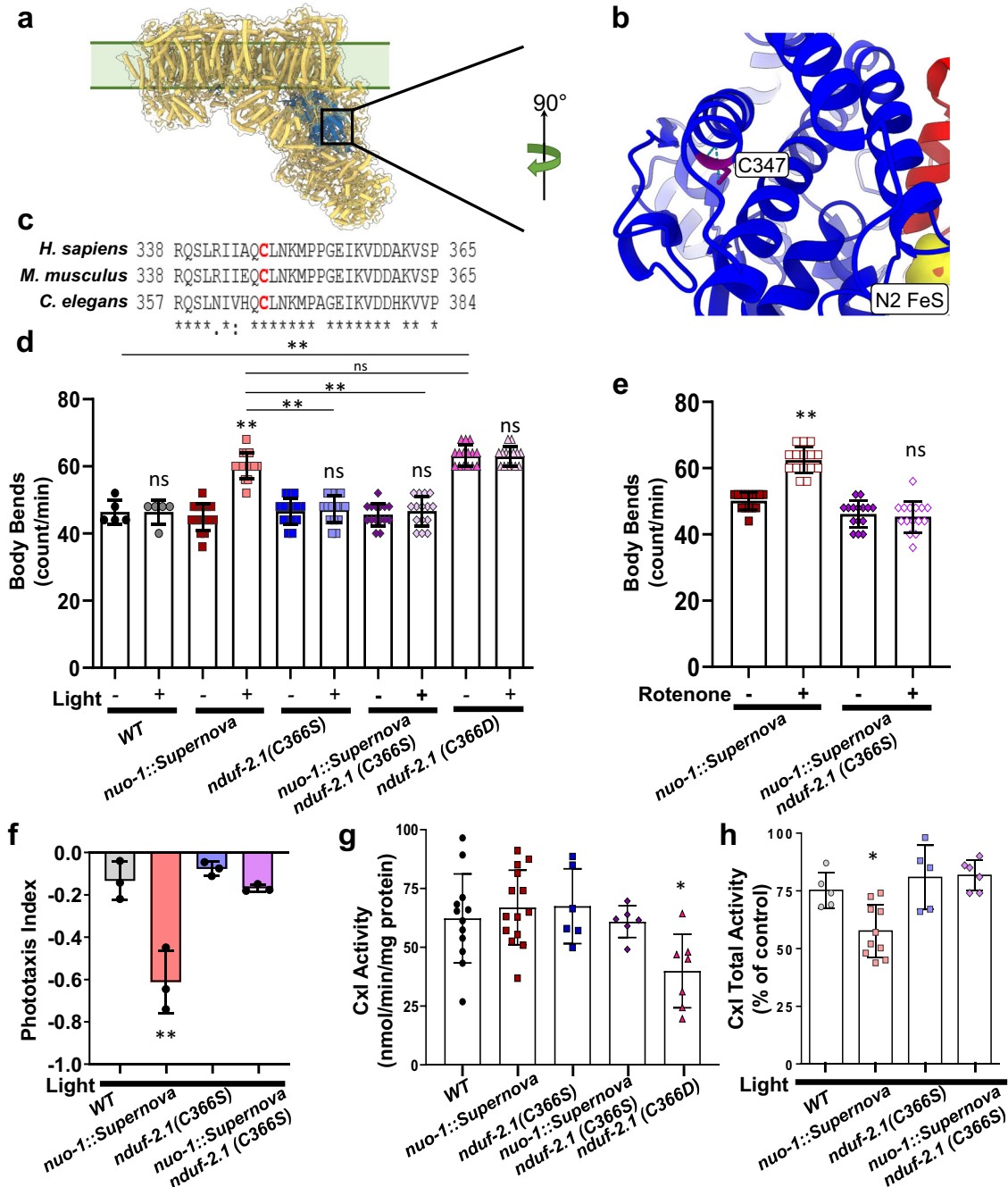

exposure to hypoxia, the Cys366Asp mutant rapidly decreased locomotion and could not respond to changes, suggesting that reversible modifications at this site regulate dynamic changes in behavior (Fig. 7a). The Cys366Asp finding is in agreement with the response of paraquat-treated worms, whereby the paraquat treatment resulted in worms not able to respond to changes oxygen concentration (Supplementary Fig. 1E).

Since both the Cys366Ser and Asp mutant could not sense acute changes in oxygen, we next tested if they are sensitive to hypoxia-reoxygenation (HR) injury and if hypoxic preconditioning can protect against HR. Hypoxic preconditioning is an endogenous protective signaling mechanism that occurs in response to non-lethal hypoxic exposures. Both the WT and nuo-1::Supernova strains were sensitive to HR and could be protected against it via hypoxic preconditioning. The Cys366Ser and Asp mutants were sensitive to HR and the degree of

protection from PC was decreased (Fig. 7a–c). The mechanism of hypoxic preconditioning involves ROS[34] and the protective effect was diminished in the Cys366 mutant strains. Therefore, we hypothesized that the light-induced reversible modification could provide protection against HR. To test this, we subjected the nuo-1::Supernova strain to HR and generated light-induced ROS at different time points. We found that light given immediately prior to the hypoxic insult was sufficient for protection (Fig. 7d, e). Overall, these results demonstrate that reversible modification of Cys366 is necessary for hypoxic signaling and that this signaling is beneficial against a lethal hypoxic insult.

## Discussion

Using a novel optogenetic fusion protein approach, combined with pharmacology and traditional genetics, we describe a model for precise mitochondria-mediated redox signaling in vivo

**Fig. 6 *nduf-2.1* Cys366 mediates complex I ROS induced increase in locomotion. a** Schematic illustration of complex I (PDB:6ZR2) showing site of NDUFS2 Cys347 (mammalian ortholog of *C. elegans* NDUF-2.1 Cys366). **b** Expanded crystal structure of mammalian C347 of NDUFS2 (blue) and nearby NDUFS7 (red). **c** Ortholog alignment of *H. sapiens*, *M. musculus* and *C. elegans* position of the Cys residue. **d** Complex I ROS induced increase in locomotion requires *nduf-2.1* Cys366. Wild-type, *nuo-1::Supernova*, *nduf-2.1(C366S)*, *nuo-1::Supernova + nduf-2.1(C366S)*, and *nduf-2.1(C366D)* L4 worms were singled and transferred to an unseeded plate. Body bends were scored for 15 s with and without illumination (MVX, 5.6 mW/mm$^2$). Data are mean ± SD. $N = 15$ independent animals across 3 technical replicates, **$p < 0.01$ (Two-way ANOVA, Tukey's multiple comparisons). **e** Rotenone increase in *C. elegans* locomotion requires *nduf-2.1* Cys366. *nuo-1::Supernova* and *nuo-1::Supernova + nduf-2.1(C366S)* were treated with 1 µM rotenone for 24 h. Body bends were scored on unseeded plates. Data are mean ± SD. $N = 15$ independent animals across 3 technical replicates, **$p < 0.01$ (Two-way ANOVA, Sidak's). Full dose response in Supplementary Fig. 1. **f** Complex I ROS-mediated phototaxis requires *nduf-2.1* Cys366. Wild-type, *nuo-1::Supernova*, *nduf-2.1(C366S)*, and *nuo-1::Supernova with nduf-2.1(C366S)* worms were transferred to the center of the seeded plate with half of the plate shaded (dark). Worms were then illuminated (GYX, 0.78 mW/mm$^2$) for 120 min and the number of worms in light and dark sections were scored and the phototaxis index was calculated. Data are mean ± SD. $N = 3$ independent experiments, each containing 40–90 animals, **$p < 0.01$ (Two-way ANOVA, Tukey's multiple comparisons). **g** *nduf-2.1* Cys366Asp results in loss of complex I activity. Total complex I activity was measured from wild-type, *nuo-1::Supernova*, *nduf-2.1(C366S)*, *nuo-1::Supernova with nduf-2.1(C366S)*, and *nduf-2.1(C366D)* isolated freeze-thawed mitochondria. Data are mean ± SD. $N = 12, 14, 6, 6, 7$ independent mitochondrial preparations, *$p < 0.05$ vs WT (One-way ANOVA, Dunnett's multiple comparisons). **h** Light-induced complex I inhibition requires *nduf-2.1* Cys366. Isolated freeze-thawed mitochondria from wild-type, *nuo-1::Supernova*, *nduf-2.1(C366S)*, and *nuo-1::Supernova + nduf-2.1(C366S)* worms were exposed to light (GYX, 7.8 mW/mm$^2$) for 10 min and total complex I activity was measured and expressed as percent of no light control. Data are mean ± SD. $N = 5, 10, 5, 6$ independent mitochondrial preparations, *$p < 0.05$ vs all groups (One-way ANOVA, Tukey's multiple comparisons).

(Fig. 8). We implicate a single thiol residue in mitochondrial complex I (Cys366) in mediating hypoxic behavioral responses due to reversible oxidation by compartmentalized ROS (hydrogen peroxide). How the redox-mediated avoidance is communicated from mitochondria throughout an organism is unclear. Hydrogen peroxide is a known signaling molecule with partially characterized function in *C. elegans* neurons, whereby hydrogen peroxide activates peroxiredoxin-mediated signaling cascades[4,5]. However, the effects of ROS are largely specified by the compartment in which they are generated. In addition, worms are known to respond to energy status by modulating locomotion speed through an unknown mechanism[35]. Likewise, *C. elegans* also respond to rapid changes in oxygen concentrations through a well-studied EGL-9/HIF pathway involving CYP-13A12 oxidation of unsaturated fatty acids[11,36]. Our results suggest that complex I redox modification may be a proximal sensor for this effect that could act in parallel or upstream of HIF signaling, as a necessary and sufficient signal. Overall, our model combines these lines of work through complex I ROS signaling and describes a paradigm where the complex I redox state couples oxygen status to acute behavior.

We found that Cys366 oxidation by hydrogen peroxide was required for the behavioral avoidance response. Complex I is known to produce ROS when there is a high NADH/NAD$^+$ and coenzyme QH$_2$/Q ratio[27], conditions where electron entry is not limiting energy production. Similarly, hypoxia increases these ratios[25,27]. The coordination between nutrient sensing, metabolism, and redox signaling is complicated, and identifying their selective physiologic effects is confounded by their intersecting roles as both process mediators and dependent factors.

The hypoxic avoidance response occurs rapidly and is reversible. This is fundamentally different from other hypoxic signaling paradigms, where complex I is modified and degraded in an attempt to protect against oxidative stress in ischemia reperfusion injury[24]. Conversely, the response we observe seems to act as a signal of energy imbalance, rather than a response to energy collapse that is already underway. The reversibility and plasticity of the response support this model of an alarm signal rather than a damage signal.

In addition, the photo-locomotive response depends on the topology of complex I within the ETC, with ROS generators, scavengers, and reactive protein residues being situated in a functional microdomain. Hydrogen peroxide can modify thiols and through its physical characteristics and ability to cross lipid membranes is generally thought to link local ROS production to

more distal modifications[37,38]. The spatial selectivity of the behavioral response to complex I ROS suggests that the local production of superoxide was sufficient to restrict ROS microdomain. However, the overproduction of ROS at any site may eventually modify Cys366 as well as indiscriminately modify other proteins, DNA and lipids. We speculate that the overproduction of ROS will not result in a selective response but would rather lead to oxidative damage and death. We demonstrated that local superoxide production impacts local hydrogen peroxide levels to direct behavioral plasticity, and that *sod-2* mutant animals lack the avoidance response to complex I ROS (Fig. 5d). These findings, together with the observation that *sod-2* localizes to complex I supercomplexes[31], suggest a privileged complex I redox environment that is coordinated by ROS production and scavengers to promote behavioral homeostasis.

We demonstrate that behavioral responses can be regulated by redox modification of a single cysteine residue in NDUF-2.1. The Cys366Ser and Cys366Asp mutants are not identical to a redox modified cysteine however, the mutants provide a binary response in activity to test cause and effect of the redox-active residue. The NDUF-2.1 mammalian orthologue, NDUFS2, is essential for carotid body[25] and pulmonary vascular[26] oxygen sensing. The highly conserved Cys366 residue and its orthologs were identified as redox-sensitive[29] and post-translationally modified[24,28] across mammalian model systems. Not only do these multiple lines of rationale support Cys366, but they also suggest that a similar mechanism of complex I ROS signaling may be occurring in other oxygen-sensing tissues functioning as a novel mechanism of ROS-mediated physiologic control[39,40].

Normal mode analysis suggests that steric clashes induced by redox modification of the human ortholog of Cys366 could shift the rigid NDUFS2 helical body, leading to an expansion of the Q binding pocket and subsequent Q instability. Mutations to residues directly adjacent to the Q pocket, which putatively stabilize this region, are already known to disrupt complex I activity[41,42]. Here, we show that the reversible oxidation of residues distant from the Q binding pocket can also disrupt complex I activity, supporting the role of reversible, site-specific oxidation events in controlling metabolism. Together, our findings recapitulate the effects of hypoxia in other oxygen-sensing tissues and provide a molecular mechanism to support the role for complex I ROS in oxygen sensing.

Finally, existing approaches to study complex I ROS through genetic or pharmacologic interventions lack selectivity and reversibility, which, given the spatial and temporal precision of

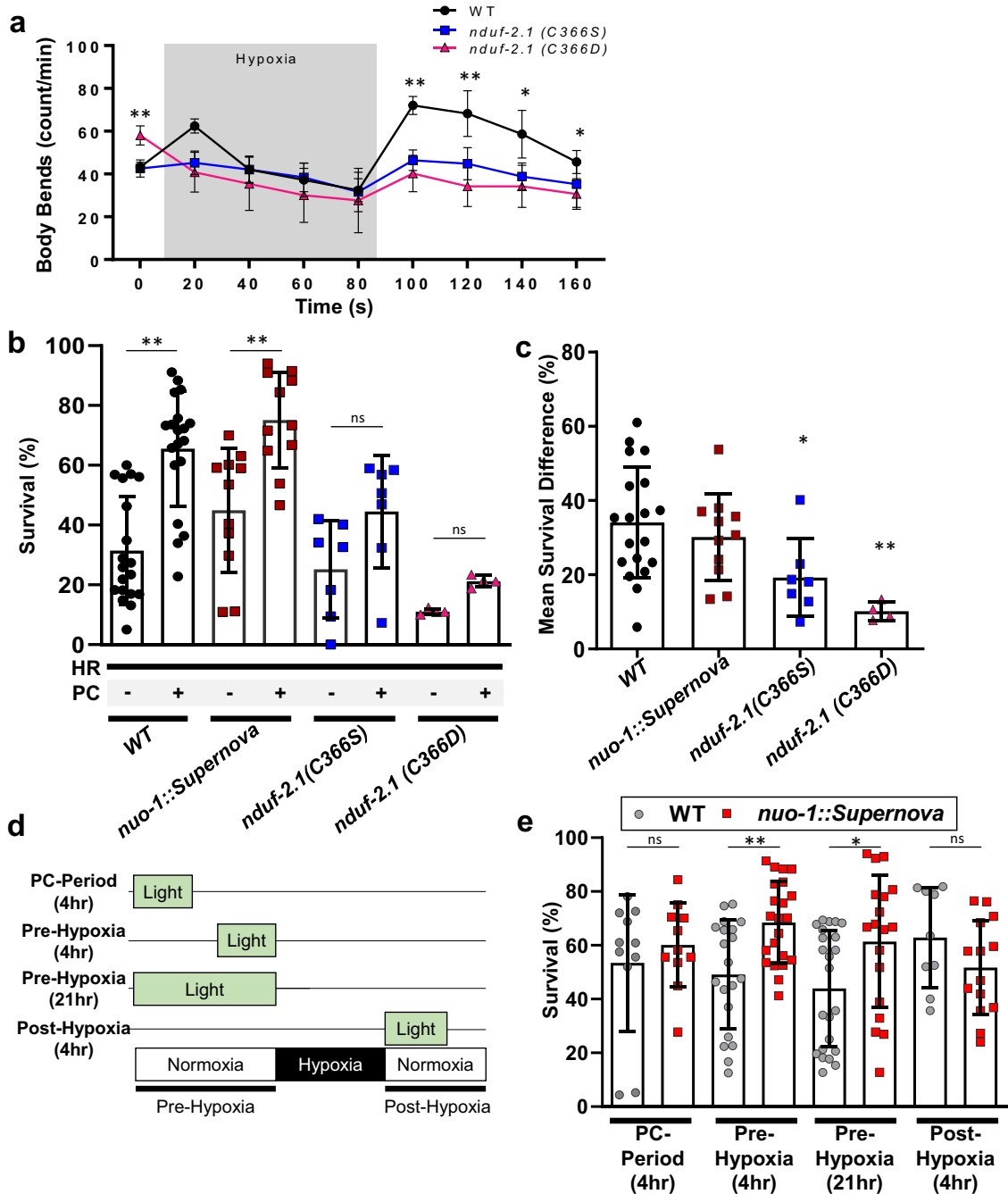

**Fig. 7 nduf-2.1 Cys366 effects on hypoxic signaling. a** Complex I ROS requires Cys366 for acute hypoxic signaling. Staged L4 wild-type, n*duf-2.1(C366S)*, *nduf-2.1(C366D)* worms were singled on to an unseeded plate and baseline body bends were counted. Worms were then subjected to a short period of hypoxia and subsequent reoxygenation. Data are mean ± SD. $N = 15$ independent animals across 3 technical replicates, $*p < 0.05$, $**p < 0.01$ (Two-way RM ANOVA, Dunnett's multiple comparisons). **b** Complex I ROS requires Cys366 for hypoxic signaling and protection against hypoxia reperfusion injury. Staged L4 wild-type, *nuo-1::Supernova, nduf-2.1(C366S)*, *nduf-2.1 (C366D)* worms were subjected to hypoxia reperfusion injury and survival was assessed. Where indicated, worms were exposed to a hypoxic preconditioning stimulus. Data are mean ± SD. $N = 4$–19 independent replicates each containing 25–100 animals $**p < 0.01$ (two-way ANOVA, Sidak's multiple comparisons). **c** Mean survival difference was calculated from data in (**b**). Mean survival difference was calculated by subtracting the survival from hypoxia reoxygenation group from the preconditioning group for each daily technical replicate, Data are mean ± SD. $N = 4$–19 independent replicates, $*p < 0.05$, $**p < 0.01$ (one-way ANOVA, Dunnett multiple comparisons). **d** Schematic illustration of different light treatments pre- and post-hypoxic conditions. **e** Complex I ROS is sufficient to protect against hypoxic injury. Stage L4 wild-type, *nuo-1::Supernova* worms were exposed to light (Quantum, 0.02 mW/mm²) as indicated in (**d**). Light was given for 4 h during the preconditioning period (PC-period), 4 h immediately prior to the hypoxic insult, 21 h immediately prior to the hypoxic insult, or 4 h immediate after the hypoxic insult. Survival was scored 24 h post-reoxygenation. Data are mean ± SD. $N = 9$–23 independent replicates each containing 25–100 animals, $*p < 0.05$, $**p < 0.01$ (Two-way ANOVA, Sidak's multiple comparisons).

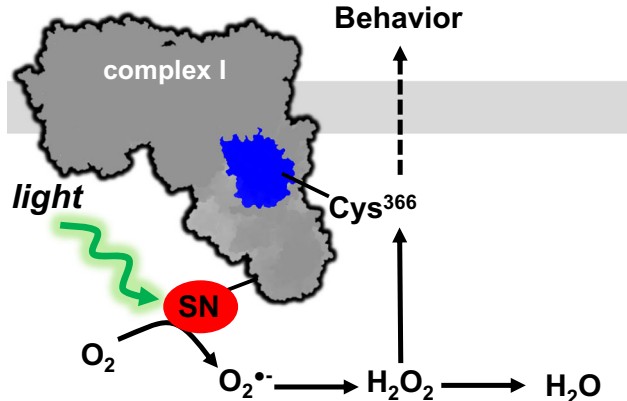

**Fig. 8** Proposed mechanism of mitochondrial complex I ROS mediating hypoxic avoidance behavior.

the ROS signaling presented has obvious drawbacks. For example, *C. elegans* complex I mutants make ROS and are useful experimental models, but they also have slow growth rate, small brood size, and lack of ETC supercomplexes[24,43–45]. Deciphering when, where, and how much ROS is sufficient to drive these phenotypes in mutant strains can be challenging and further complicated by potential adaptive feedback mechanisms[46]. Likewise, detecting localized ROS can prove to be challenging and measures of ROS are not without caveats. The optogenetic isolation of complex I ROS independent of confounding factors present in this study has enabled mechanistic investigation to complement these pharmacologic and genetic tools.

In conclusion, we propose a model of redox signaling where mitochondrial complex I acts as a redox rheostat to couple the levels of ROS (damaging versus signaling amounts) with environmental oxygen availability. This redox-energetic coupling can serve either to decrease damage during a heart attack or stroke as is well described by the literature, or as we report here, it can serve to alert an organism to impeding damage and trigger avoidance.

## Methods

**Worm strains and maintenance**. *C. elegans* were maintained at 20 °C on plates containing nematode growth media (NGM) with OP50 bacterial as food[47] Strains used were generated in-house or provided through the *C. elegans* Genetics Center (CGC) and are listed in Supplementary Table 1. This study used L4, Day 1, Day 3, and Day 5 adult hermaphrodites as indicated in figure legends. *C. elegans* strains and plasmids are available upon reasonable request or via the Caenorhabditis Genetics Center. Experiments were approved by the University of Rochester Institutional Biosafety Committee.

**Transgenic *C. elegans***. CRISPR/Cas9 was used as previously described[18] to fuse Supernova to C'-terminus of *nuo-1*. Briefly, Supernova was PCR amplified and inserted at the C-terminus of *nuo-1* using homology-directed repair as described by Paix et al.[48]. Adult *C. elegans* gonads were injected with a mix containing 25 mM KCl, 7.5 mM HEPES, 4 µg/µL tracrRNA, 0.8 µg/µL *nuo-1* crRNA, 0.8 µg/µL *dpy-10* crRNA, 50 ng/µL *dpy-10* ssODN, 2.5 µg/µL purified Cas9, and 500 ng/µL of the supernova repair template. Three days post-injection, progeny were screened for the *dpy* phenotype and for red fluorescence. The Supernova edit was sequenced, and the strain was outcrossed with wild-type worms in order to remove the *dpy* phenotype. Other electron transport chain fusions constructs were generated using the same procedure. Point mutations to *nduf-2.1* were generated using this process except using a *nduf-2.1* crRNA (0.8 µg/µL) and single stranded oligo nucleotide (ssODN) repair template (100 ng/µL). The resulting ssODN repair resulted in a new MseI restriction enzyme site, which was used for screening. Supplementary Tables 2, 3 detail the primers and crRNA sequences.

**Mitochondrial isolation**. Mitochondria were isolated from staged adult *C. elegans* using differential centrifugation as previously described[18,35,49,50]. Briefly, approximately one million day 1 adult hermaphrodites were grown on HB101. *C. elegans* were rinsed with M9 media (22 mM KH₂PO₄, 42 mM Na₂HPO₄, 86 mM NaCl, 1 mM MgSO₄, pH 7) and placed in mitochondrial isolation media (220 mM

mannitol, 70 mM sucrose, 5 mM MOPS, 2 mM EGTA, pH 7.3 at 4 °C). *C. elegans* were then homogenized using pure sea sand in an ice-cold mortar followed by Dounce homogenization. Mitochondria were enriched from the homogenate through differential centrifugation using mitochondrial respiratory media (220 mM mannitol, 70 mM sucrose, 5 mM MOPS, 2 mM EGTA, 0.04% BSA, pH 7.3 at 4 °C). Protein concentration was determined using the Folin-phenol method[51].

**Lighting systems**. Illumination systems include: a light-emitting diode (LED) system (abbreviated as "GYX", GYX module, 540–600 nm, X-Cite LED1; Excelitas, Waltham MA), an X-Cite 220 V mercury bulb (abbreviated as "MVX", 540-580 nm Texas Red excitation filter, Excelitas, Waltham MA) and a SpectraLife LED array (abbreviated as "Quantum", 580–600 nm, Quantum Devices, Barneveld, WI). Prior to illumination, the lighting system was calibrated with an optical power meter (1916-R, Newport Corporation) and thermopile detector (818P-010-12; Newport Corporation, Irvine, CA). Lighting system and intensities are listed for each experiment.

**Mitochondrial respiration**. Mitochondrial respiration was measured using Clark-type O₂ electrode (Hansatech Instruments, UK)[49,52]. Briefly, following calibration of the electrode, 1 mg/ml protein of mitochondria were loaded to the chamber containing mitochondrial respiration buffer (120 mM KCl, 25 mM sucrose, 5 mM MgCl₂, 5 mM KH₂PO₄, 1 mM EGTA, 10 mM HEPES pH 7.3). Where indicated, mitochondria were illuminated (GYX, 7.8 mW/mm²) for the indicated time then subjected to respiratory analysis. Respiratory substrates (complex I-linked, 2.5 mM malate plus 5 mM glutamate; complex II-linked, 5 mM succinate with rotenone), ADP (0.4 mM), oligomycin (1 µg/ml), or rotenone (2 µM) were added to the chamber as indicated via a syringe port. Following an addition of complex-linked substrates and ADP, the maximum respiration (state 3) was determined. State 4 respiration was measured after the addition of oligomycin. Complex I RCR measures the ratio of state 3 over state 4 respirations.

**Mitochondrial enzyme activity**. Isolated mitochondria (1 mg/ml) were freeze-thawed, illuminated (GYX, 7.8 mW/mm²) and enzymatic activity was then assessed spectrophotometrically. Complex I total activity was measured as the rotenone-sensitive rate of NADH oxidation with extinction coefficient of 6180 M⁻¹ at 340 nm[49]. For activity experiments including NAC, mitochondria were first illuminated and then complex I activity was measured in the presence of NAC. Complex I NADH dehydrogenase activity was determined as NADH driven, diphenyleneiodonium rate of DCPIP reduction with extinction coefficient of 21000 M⁻¹ at 600 nm[53,54] or hexaammineruthenium sensitive rate of NADH oxidation[55,56]. Complex II was calculated as the malonate-sensitive succinate oxidation rate of DCPIP reduction with extinction coefficient of 21000 M⁻¹ at 600 nm[18,49]. Citrate synthase activity was assessed as the rate of DTNB-coenzyme A formation with extinction coefficient of 13600 M⁻¹ at 412 nm[18,49].

**Clear native electrophoresis**. Isolated mitochondria were illuminated (GYX, 0 and 7.8 mW/mm²) and clear native electrophoresis was performed as published[57]. Briefly, 20 µg of mitochondrial protein was solubilized with digitonin (6 µg digitonin/µg protein) for 20 min to maintain supercomplexes. Loading buffer (1/10 v/v) was added and proteins were separated on a 4–10 % polyacrylamide gradient gel at 4 °C. Gels were subjected to a complex I in-gel assay[57,58], stained with Coomassie or transferred onto nitrocellulose membranes (Trans-Blot Turbo Transfer system, BioRad). Membranes were stained with Ponceau S to visualize protein loading, blocked with 5% milk in TBST, incubated with primary antibodies (rabbit anti-KillerRed, Evrogen AB961; Cat#AB961, Lot#96101240513; 1:1,000 dilution; Note Supernova is the monomeric version of tdKillerRed) followed by fluorescent labeled secondary antibodies (Goat-anti-rabbit, Starbright 700, BioRad; Cat# 12004158, Lot#64247470, 1:5,000 dilution). Signals were detected using a ChemiDoc station (BioRad) and images were quantified using ImageJ.

**Superoxide measurement**. Superoxide was measured using 2-hydroxyethidium (2-OHE⁺), the superoxide-selective dihydroethidium (DHE) oxidation production, as previously described[14,15,18]. Freshly isolated mitochondria were illuminated (GYX, 7.8 mW/mm²) in the presence of DHE (100 µM). Protein was precipitated using 200 mM HClO₄/MeOH and removed via centrifugation. An equal volume of phosphate buffer (1 M, pH 2.6) was added to the supernatant. The resulting samples were filtered and separated using a polar-RP column (Phenomenex, 150 × 2 mm; 4 µm) on an HPLC (Shimadzu) with fluorescence detection (RF-20A). The protocol consisted of two mobile phases (A: 10% ACN, 0.1 % TFA; B: 60% ACN, 0.1 % TFA) with the following gradient: 0 min, 40% B, 5 min, 40% B; 25 min, 100% B; 30 min, 100% B; 35 min 40% B; 40 min, 40% B. A standard curve was generated using purified 2-OHE⁺ and samples were quantified using Lab Solutions (Shimadzu).

**HyPer7 imaging**. Staged L4 worms were grown on standard NGM plates and screened for bright pharyngeal expression of HyPer7. Prior to imaging, worms were placed into a glass-bottom 96-well plate containing M9 buffer. Tetramisole hydrochloride 10 mM was added to each well roughly 30 min prior to imaging.

Light exposure (GYX module, 1.44 mW/mm$^2$) was induced for 2 min. Imaging was performed using a Nikon Eclipse inverted microscope coupled to a six channel LED light source (Intelligent Imaging Innovation, Denver, CO), an ORCA-Flash4.0 V2 Digital CMOS camera (Hamamatsu Photonics, Bridgewater Township, NJ). Both 400 nm and 500 nm excitation images were taken with 50 ms of exposure time and analyzed using Slidebook6 software (Intelligent Imaging Innovation, Denver, CO). All images were acquired under the same exposure conditions. Automated noise reduction was completed by removing particles less than 10 px in diameter and background was subtracted using a region of interest without signal on each image.

**Confocal imaging**. Staged adult worms were anesthetized using 0.1% tetramisole and imaged using an Olympus FV 1000 confocal microscope using a 60x oil objective. Z-stack images were smoothed by a three-point moving average of pixel intensity and normalized to maximum intensity of each profile using imageJ (1.5.2)[18].

**Brood size assay**. Individual staged L4 worms were moved to OP50-seeded plates every 24 h for 8 days. The viable progeny from each plate were counted daily to determine the brood size.

**Locomotion assays**. Staged worms were moved individually to an unseeded plate. Worms were equilibrated for 30–60 s. Body bends were counted for 15 s without light to collect the baseline and then illuminated (MVX, 5.6 mW/mm$^2$). For measurements involving drugs or toxins, L4 worms were transferred to seeded plates containing the drug or toxin for 24 h prior to the experiment. For short-term hypoxia studies, staged L4 worms were acclimated to an unseeded plate for 30–60 s. The plate was then inserted into a hypoxic chamber (<0.01% O$_2$, 5%H$_2$/95%N$_2$, 26 °C, palladium catalyst, Coy Laboratory Products) and body bends were immediately counted every 20 s for 70 s. Following 70 s in hypoxia, plates were returned to normoxia and body bends were counted for an additional 60 s. For the phototaxis assay, staged L4 worms (25–80) were placed in the center of a seeded plate with half of the plate shaded (dark) while the other half was exposed to light (light). Worms were illuminated (GYX, 0.78 mW/mm$^2$) for 120 min after which worms in each section were scored. Phototaxis index was calculated as the number of worms in the light section minus the number in the dark divided by the total.

**Hypoxia-reoxygenation injury**. Synchronized day 1 adults (50–100 worms/plate) were placed in a hypoxic chamber (<0.01% O$_2$, 5%H$_2$/95%N$_2$, 26 °C, palladium catalyst, Coy Laboratory Products) for 18.5 h. Plates were then returned to normoxia for 24 h and survival was scored. Moving animals or animals that moved in response to a light touch were scored alive. Preconditioning protects *C. elegans* from hypoxia reperfusion injury and is modeled by exposing worms to a 4 h hypoxic stimulus 21 h prior to the hypoxic insult. Where indicated plates were illuminated (Quantum, 0.02 mW/mm$^2$) for 21 h prior, 4 h prior, or 4 h post-hypoxic insult. To mimic the preconditioning stimulus, plates were illuminated for 4 h, 21 h prior to the hypoxic insult. Mean survival difference was calculated by subtracting the survival from hypoxia reoxygenation group from the pre-conditioning group for each daily technical replicate.

**Structural analysis**. Active state human complex I (PDB: 6ZR2; [https://doi.org/10.2210/pdb6ZR2/pdb]) was loaded into ProDy (v2.0)[59] for normal mode analysis of all 8171 alpha carbons. The resulting modes were visualized using the NMWizard plugin in VMD (v1.9.4)[60]. The ubiquinone-like piericidin molecule bound to human complex I (PDB: 6ZTQ) was superimposed to approximate the Q binding pocket.

**Statistical analysis**. All the data were first subjected to a normality test and then analyzed by an unpaired, two-tailed t-test or one- or two-way ANOVA with post hoc multiple comparison correction as indicated in the figure legends. Statistical p values < 0.05 were considered significant (GraphPad Prism 7 and 9, Microsoft excel 2019).

**Reporting summary**. Further information on research design is available in the Nature Research Reporting Summary linked to this article.

## Data availability

Data generated with this study are available in the main text, and supplementary figures and tables. Previously published databases and accession codes used in the study (e.g. PDB: 6ZR2) are listed in the main text. Source data are provided with this paper.

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

## Acknowledgements

Work in the laboratory of APW is supported by grants from National Institutes of Health (R01 NS092558, R01 NS115906). Some strains were provided by the Caenorhabditis Genetics Center (CGC), which is funded by NIH Office of Research Infrastructure Programs (P40 OD010440). BJB is supported by the Biological Mechanisms for Healthy Aging (BMHA) Training Grant NIH T32AG066574. BJB current affiliation: University of Washington, Department of Laboratory Medicine & Pathology, Seattle WA, 98195, United States of America. VVB was funded by grant 075-15-2019-1789 from the Ministry of Science and Higher Education of the Russian Federation. We also thank the Mitochondrial Research & Innovation Group at University of Rochester Medical Center and the Western New York Worm Group for helpful discussions.

## Author contributions

J.O.O. and A.P.W. conceived and oversaw the project. J.O.O., B.J.B., S.A.K., and A.P.W. wrote the manuscript. J.O.O., M.A.F., G.B., G.A.P., A.V., S.A.K., A.B., B.J.B., and A.P.W. designed and performed experiments. V.V.B. provided essential reagents and HyPer7 expertise. A.G. contributed to the structural analysis. A.P.W. provided grant support. All authors edited and approved the final manuscript.

## Competing interests

The authors declare no competing interests.
