## [Peer Review File · Nature Communications]

A reversible mitochondrial complex I thiol switch mediates hypoxic avoidance behavior in *C. elegans*REVIEWER COMMENTS

Reviewer #1 (Remarks to the Author):

Overview

This is a very interesting paper that uses *C elegans* and optogenetic approaches to explore complex I signalling in response to hypoxia. The overall model suggested is very appealing and much of the data here are supportive. However, there are a few gaps and I also have some technical queries.

Major points

1 The model is based on the idea that complex I produces superoxide under hypoxia and this moves on to make the worm avoid hypoxic conditions. However, it was not clear to me how hypoxia led to an increase in superoxide production, nor how the oxidation of the cysteine on complex I leads to the change in movement. While I understand that these may not yet be understood it's important that these points are clarified.

2 The use of EUK 134 is interesting, but in all cases using specific compounds like this it is important to use controls as similar as possible but without the catalytic activity.

3 The conjugation of Supernova to nuo-1 is a very nice development. The confocal data shown are not useful for this and are not convincing. It was unclear why the authors used mCherry for this, rather than the nuo-1/supernova itself? However, it's vital to show that it is actually incorporated into intact complex I. This has to be done. I think BN-PAGE is probably the best method.

4 The measure of "body bends" with various interventions seems to show a relatively small effect with paraquat, or with light etc. But the effect on omega bends/direction changes seems relatively larger. Would it be better to focus on these measurements?

5 The effect of light on superoxide production on the supernova construct is relatively small. I guess in an ideal world the authors could link Hyper7 to complex I as well, but that's for the future. Presumably the low levels are due to the dispersal of the probes within the mitochondria compared to the local production at complex I. But often the data are scattered and the effects are small.

6 The effect on complex I activity seen in Figure 4 are very impressive. It would be good to correlate the effect on cys oxidation with that on complex I activity. Have the authors considered assessing cys 366 oxidation/modification by mass spec and correlate this with the change in activity?

7 In Figure 4F the assessment of NADH activity by DCIP is a poor assay as other enzymes can interfere. This should be repeated with NADH/hexammineruthenium.

8 In Figure 4G, the effect of NAC was unclear as it was present during the light, so the NAC may be blocking the effect, not reversing it. This could be done by blocking with light and then seeing if adding NAC afterwards reversed the effect.

9 The structural model in Fig 6 seemed very speculative to me. While complex I structure is highly conserved I did not feel the benefit of this analysis was worth the speculation – better to wait until there is a *C. elegans* complex I structure, or also consider comparisons with other complex I structures from bacteria and yeasts.

10 The effects on hypoxia-reperfusion are intriguing. However, to extrapolate this as a mechanism for preconditioning, is a bit of a stretch. There have been many, many models proposed for preconditioning so best not to over interpret these data.

Reviewer #2 (Remarks to the Author):

This is an exciting paper, that employs a mix of advanced methods, to propose that a thiol “switch” within complex I is key for hypoxia avoidance behavior in *C. elegans*. Using a CI:SuperNova fusion to enable photo-activatable superoxide formation the authors are able to achieve light-induced avoidance behavior and an increase in speed similar to that induced by hypoxia. By performing state 3/ state 4 measurements on mitochondria isolated from these worms, they observed a decrease in state 3

respiration following CI:SuperNova activation with CI linked substrates but not with CII substrates — leading to the hypothesis that CI linked respiration is somehow halted. The effect of complex I::SuperNova was abolished by treatment with ROS scavengers or loss of sod-2, further supporting a role for complex I generated ROS and a reversible modification on CI in mediating the behavioral response. Given that mammalian NDUFS2 has been known to be important in hypoxia signaling, the authors considered three conserved NDUFS2 residues recently reported to be oxidized in a proteomic screen of oxidized cysteines. They focus on Cys366 and create a serine (incapable of getting oxidized) or aspartate (mimicking oxidation) mutant to demonstrate that oxidation of this residue is necessary and sufficient to mediate the behavioral response to SuperNova and hypoxia. They end by showing that oxidation of this residue also mediates ischemic preconditioning. Although some elements of the core concepts were known in mammalian systems (reversible oxidation of CI cysteines in the context of ischemia reperfusion injury) it is an overall exciting story.

Major critiques:

1. My major concern is related to the purported specificity of Complex I ROS in inducing this behavior. Many perturbations that make worms sick will induce an avoidance response. In fact, the authors do attempt this control, by tethering SuperNova to two distinct places on Complex II. They conclude that the behavioral effect was not recapitulated with complex II:Supernova (lines 132-139). However, in contrast to what is written in the text, the data in the figure (Figure S4) *do* suggest that both of these Complex II constructs have an effect. *sdhc-1::SuperNova* significantly increases body bends in a light-dependent manner, and *sdhb-1::SuperNova* increases body bends at baseline to above that achieved by *nuo-1::SuperNova* activation. If the authors want to claim that ROS generation at Complex I is unique, they need to address this inconsistency.
2. I am also concerned about the purported specificity of the C366 residue in mediating the behavioral response to Complex I ROS and hypoxia. Does the C366S mutation block the drop in CI State 3 activity from complex I::SuperNova? Can the authors test other available Complex I mutants to see if they affect the behavioral response to hypoxia or Complex I ROS (as in Figs 7A and 6D, respectively). The *gas-1(fc21)* strain is already used in this study, for example.
3. The primary phenotype (behavioral response to hypoxia and re-oxygenation) has previously been investigated by the Horvitz lab (PMID: 22405203 and 23811225). These studies identified many genes in the EGL-9/HIF-1 pathway that control this behavior. Can the authors use genetics place their findings in the context of this pathway, e.g., are EGL-9 and CYP-13A12 are required for the behavioral response to Complex I:Supernova? At the very least the authors should discuss in length their work in the context of this important body of work from the Horvitz lab.

4. I found the elastic network modeling very difficult to understand and distracting. This modeling is highly speculative, not needed, and detracts from an otherwise compelling story. I'd urge the authors to consider removing this part of the paper.

Reviewer #3 (Remarks to the Author):

In this manuscript, Onukwufor and colleagues explore the mechanism underlying avoidance behaviors, under hypoxic conditions in *C. elegans*. Triggered by previous reports, which suggest an association between mitochondrial complex I ROS production and hypoxia, the authors tested whether mitochondrial complex I is involved in a specific locomotory response upon hypoxia and reoxygenation. To monitor ROS production in vivo, they created transgenic animals expressing the optogenetic ROS-generating protein SuperNova fused to an endogenous site of ROS production, *nuo-1*, by using the CRISPR-Cas9 system. They found that animals expressing the *nuo-1::SuperNova* sensor showed increased forward movement upon photo-activation as well as increased direction changes compared to wild-type worms. The authors found that the activation of avoidance through *nuo-1::SuperNova* photoactivation was abolished in *sod-2* or/and *sod-3* knockout strains, suggesting that complex I ROS mediated an increase in locomotion. Conversely, the addition of a SOD mimetic exacerbated light-induced ROS effects and rescued the photo-locomotion response in the absence of SOD. Furthermore, by using a superoxide and catalase mimetic EUK-134 and NAC, the authors obtained similar results since hydrogen peroxide was required for the activation of avoidance upon *nuo-1::SuperNova* activation. Then, the authors focused on the *nduf-2.1* gene (the *C. elegans* orthologue to mammalian NDUFS2 subunit 2), which forms part of the mammalian coenzyme Q-binding site in complex I. In particular, they provide evidence suggesting that Cys366 was both necessary and sufficient to mediate ROS-induced behavioral changes. The Cys366Ser and Asp mutants could not sense acute changes in oxygen; they were also sensitive to hypoxia re-oxygenation (HR) and the degree of protection from hypoxic preconditioning (PC) was decreased in these mutant animals. The study suggests that a single thiol residue (Cys366) in the NUDF-2.1 subunit of mitochondrial complex I mediates hypoxic behavioral responses due to reversible oxidation by compartmentalized ROS. As a consequence, complex I enzymatic activity is decreased probably because of destabilization of the coenzyme Q-binding pocket. The manuscript presents some potentially interesting findings and tools. Implementation of a novel optogenetic fusion protein approach to model mitochondria-mediated redox signaling in vivo is an added plus. However, key claims made by the authors are not sufficiently supported by data in the manuscript.

The authors suggest that their findings (as mentioned above) support a direct role of complex I ROS in mediating hypoxic signaling events. A rigorous support for this notion could come from the use of a *mev-1* mutant strain. *mev-1* encodes the succinate dehydrogenase cytochrome b, a component of complex II, which does not participate in the NADH to oxygen redox/ proton pumping cascade and thus it would not be expected to generate ROS under normal redox conditions.

Also, with respect to the regulation of hypoxic avoidance behavior by complex I ROS, the authors suggest that this idea is further supported by the fact that complex II::SuperNova photo-activation failed to produce a behavioral response similar to that produced by complex I::SuperNova photo-activation. To strengthen their statement, they should test the behavior of animals expressing a complex III::SuperNova transgene since complex I and III are the main sites of ROS production in mitochondria.

In figure 1A, a superoxide dismutase and catalase mimetic EUK-134 renders nematodes unresponsive to changes in oxygen levels, presumably by sequestering ROS. However, the reverse experiments using paraquat and rotenone to elevate mitochondrial ROS production have been conducted under baseline conditions (figure 1B-C). The authors should specifically address whether higher ROS are accompanied by increased motility when animals face a hypoxia-reoxygenation stress and perform a longitudinal locomotion analysis, similarly to figure 1A.

In figure 2, following an HPLC-based approach, the authors validate that nuo-1::Supernova transgenic can generate excessive ROS upon illumination with a light source. They can use additional methods to corroborate this finding. For example, they can measure ROS in worm lysates using CM-H2DCFDA (PMID: 29707606) or utilize fluorescent dyes (such as Mitotracker ROS) to quantify mtROS in vivo. Better quality and higher magnification images would be more informative.

In Figure 2B-C, the authors are using Paraquat and rotenone in order to investigate whether ROS originating from complex I cause changes in locomotion behavior. They should show whether these drugs lead to the same or additive effects regarding locomotion under hypoxic conditions.

In Supplementary Fig. 4C, the authors investigate whether light exposure of complexII::supernova induces similar effects to those of complex I::supernova. At least for the sdhc-1::supernova, it seems that mild light exposure increases body bends. Therefore, it could be that ROS generation from complex II may also contribute to the behavioral change. The fact that there is not dose-dependent effect in higher concentrations is adequate to exclude complex II contribution? Do the authors have any indication for the involvement of other ETC complexes except for complex I and II?

ROS levels have been associated with lifespan and *C. elegans* mutants for nduf-2.1 (a.k.a gas-1) are short lived. The authors have generated a redox-insensitive C366S mutant. It would be interesting to compare the lifespan of those mutant animals, to that of the sulfinic acid modification mimetic (C366D) mutants, which phenocopies nduf-2.1 depletion. Could the identity of the C366 residue determine lifespan?

The authors have not at all discussed the widely accepted phenomenon of mitohormesis, which actually represents the positive signaling role of mild ROS production from mitochondria upon caloric restriction. Specifically, it has been shown that caloric restriction induces a transient ROS signal that improves stress

resistance leading to increased mitochondrial bioenergetics (PMID 17908557, 21619928), contrary to what is shown here (ROS production decrease mitochondrial bioenergetics).

The authors show that light affects ROS production even in the absence of SuperNova (Figure S3D). However, light does not induce an increase in body bends in wt worms (Figure S4C)? On the other hand, illumination almost doubles survival, even when imposed hours before hypoxic treatment. These findings question the relevance of the body bend assay used in this study.

The authors show that it is hydrogen peroxide generated from superoxide anion from the Supernova reporter fused to NUO-1 that affects the redox state of the cysteine366 of NDUFS2, as SOD-2 or SOD-3 depletion reverses the effects of light. However, if it is not the spatial proximity of generated superoxide anions that is responsible for Cys366 oxidation, then one would expect that superoxide anions generated in the matrix from the SN fusion to complex II would also have the same effects (increased hydrogen peroxide production and Cys366 oxidation). However, this is not the case, as SN fused to complex II failed to induce similar phenotypic effects. Why is this?

It is known that C347 oxidation status affects Complex I protein stability in mammals (PMID 27052170). Does SN activation affect protein levels of Complex I? Could this be the reason for reduced Complex I oxygen consumption? What are the effects of Complex II-fused SN on complex I stability? Western blot analysis is required.

The authors state that they identify a reversible mitochondrial complex I thiol switch. However reversibility is poorly shown. They only show that locomotion returns to normal just within a few seconds without light (Figure 3H). Based on this finding the authors suggest that thiol oxidation is reversed and ROS/H₂O₂ production as well but this is not shown. The conclusion that: "Collectively, these results suggest a reversible redox modification in complex I downstream of the NADH-binding site" is an overstatement not supported by the data shown here.

Minor comments

It would interesting to check whether complex II ROS production from SDHC-1::Supernova and SDHB-1::Supernova will exhibit increased phototaxis avoidance behaviors similar to NUO-1::Supernova upon light titration.

How does this redox modification to a single cysteine residue in NDUF2.1 affect other types of stress responses?

For the mitochondrial bioenergetic studies, it is not clear whether illumination persists throughout the oxygen consumption assay or whether it precedes it. In general, the authors need to provide a more detailed description of the protocols and procedures used.

The authors have to describe in a more concise and comprehensive way which transgenic animals they are using. For instance, in the legend of Figure 2D they refer to *sdhc-1::mCherry*, while in the results section of the main text, the particular transgene is not mentioned at all.

In figure 2B, the magnified images are blurry and should be replaced with images of higher quality.

In the graph of figure 2E, “ $p=0.12$ ” should be changed to “ns”.

Supplementary Figures should be also mentioned in the main text. For instance, Supplementary Fig. 1 is only mentioned in the figure legend of Figure 1. Several figures do not appear in text according to their numerical order.

Sup. Fig. 1: Paraquat description is missing in the legend, and panel order is scrambled.

The graphs A-D in supplementary Figure 3 should be properly aligned.

Figure 4: no annotation of time on x-axis

Figure S7C is not clear

Proper nomenclature for gene names, mutant strains, etc., should be observed throughout.

NCOMMS-21-17188-T

A reversible mitochondrial complex I thiol switch mediates hypoxic avoidance behavior in *C. elegans*

We thank the editor and reviewers for their enthusiasm and critiques, as well as for the opportunity to revise our manuscript. We have reproduced the reviewer comments verbatim below in bold and provided responses to individual points in regular typeface. Articles referenced in our response are listed using PubMed IDs (PMID). We are hopeful that the reviewers will recognize our sincere attempt to address their comments and agree that the manuscript is stronger as a result.

Reviewer #1 (Remarks to the Author):

Overview

This is a very interesting paper that uses *C. elegans* and optogenetic approaches to explore complex I signalling in response to hypoxia. The overall model suggested is very appealing and much of the data here are supportive. However, there are a few gaps and I also have some technical queries.

Major points

1. The model is based on the idea that complex I produces superoxide under hypoxia and this moves on to make the worm avoid hypoxic conditions. However, it was not clear to me how hypoxia led to an increase in superoxide production, nor how the oxidation of the cysteine on complex I leads to the change in movement. While I understand that these may not yet be understood it's important that these points are clarified.

Response: Amongst the various conditions that enhance complex I ROS formation, highly reduced NADH and coenzyme Q pools can lead to complex I superoxide production. There is evidence that hypoxia increases mitochondrial ROS production (reviewed in PMID:19061483) and we propose that short term hypoxia will cause the NADH and Q pools to become reduced, thus generating more superoxide at complex I.

How the oxidation of a single cysteine on complex I leads to behavioral changes is currently unknown and we are actively investigating this redox signaling pathway.

We expanded our discussion on how complex I generates superoxide during short term hypoxia (Discussion, lines 321-323). We have also included a discussion on the downstream mechanisms that link ROS production to whole animal behavioral changes (Discussion, lines 308-317).

2. The use of EUK 134 is interesting, but in all cases using specific compounds like this it is important to use controls as similar as possible but without the catalytic activity.

Response: We are unaware of any non-catalytic analogs of EUK-134. EUK-8 is structurally similar but has reduced catalytic activity compared to EUK-134 (PMID: 9435181). We found that EUK-8 was unable to reverse the light-induced ROS-mediated increase in locomotion (Supplemental Figure 10). Together with wildtype *C. elegans* control groups treated with these compounds (Figure 5, Supplemental Figure 10), the results suggest that the catalytic activity of the compounds rather than potential non-catalytic or off-target effects are modulating the behavioral response.

3. The conjugation of Supernova to nuo-1 is a very nice development. The confocal data shown are not useful for this and are not convincing. It was unclear why the authors used mCherry for this, rather than the nuo-1/supernova itself? However, it's vital to show that it is actually incorporated into intact complex I. This has to be done. I think BN-PAGE is probably the best method.

Response: Our goal was to test if the fusion to NUO-1 affected localization or activity of the endogenous protein. In addition to the evidence iterated below in points 1 and 2, we have added several new lines of evidence (see points 3 and 4 below) as suggested by the reviewer to provide further support for the conclusion that NUO-1::Supernova is correctly localized and functional.

1) *nuo-1* encodes the FMN-containing subunit (mammalian ortholog, *NDUFV1*). The loss of *nuo-1* is lethal and if the fusion was not functional or not incorporated into complex I, the animal would not survive. In the dark, animals expressing *nuo-1::Supernova* are phenotypically normal (Figures 2-4).

2) Isolated mitochondrial studies demonstrate that in the absence of light, complex I activity and complex I linked respiration are not affected by the fusion (Figure 4, Supplemental Figure 8). Additionally, our results show a selective light-induced loss of complex I activity (Figure 4).

3) Confocal images support mitochondrial matrix localization of the NUO-1 fusion, consistent with complex I localization. Initially, we sought to avoid potential light-induced ROS effects on mitochondrial morphology by using *nuo-1::Cherry* since Cherry and Supernova are similar in size and characteristics (28 kDa vs 29.8 kDa). However, as suggested, we repeated these experiments using *nuo-1::Supernova* and the results support earlier conclusions showing that the fusion protein is localized to the mitochondrial matrix (Figure 2, Supplemental Figure 2).

4) High-resolution clear native electrophoresis (hrCNE, PMID: 17426019) demonstrates colocalization of Supernova with complex I supercomplexes and in-gel activity assays demonstrate the complex I is functional (Supplemental Figure 3). These results, including those from experiments suggested by the reviewer, further support our conclusion that *nuo-1::Supernova* is incorporated into complex I.

4. The measure of “body bends” with various interventions seems to show a relatively small effect with paraquat, or with light etc. But the effect on omega bends/direction changes seems relatively larger. Would it be better to focus on these measurements?

Response: We acknowledge that the dynamic range of omega turns/direction changes appear larger. However, the throughput for body bend measurements (i.e., speed) is much faster and is classically used in the *C. elegans* literature PMID: 22405203. Moreover, in our opinion, subtle effects can reflect physiologically significant impacts, so long as they are reproducible, they may more fully recapitulate native regulatory pathways.

5. The effect of light on superoxide production on the supernova construct is relatively small. I guess in an ideal world the authors could link Hyper7 to complex I as well, but that's for the future. Presumably the low levels are due to the dispersal of the probes within

the mitochondria compared to the local production at complex I. But often the data are scattered and the effects are small.

Response: We are glad that the reviewer appreciates that localized ROS may have a limited impact on a diffuse indicator. The goal of these experiments was to test if our *Supernova* construct can generate superoxide in vivo, and the results, albeit modest, support this idea. We think that it is important to recognize too that large scale ROS production is often associated with oxidative damage while localized ROS production can act as signaling events.

6. The effect on complex I activity seen in Figure 4 are very impressive. It would be good to correlate the effect on cys oxidation with that on complex I activity. Have the authors considered assessing cys 366 oxidation/modification by mass spec and correlate this with the change in activity?

Response: Methods to quantify reversible redox modifications via mass spec are rapidly advancing, but are outside our current expertise. Instead of correlational measures, we sought to directly test the role of Cys366 through CRISPR/Cas9 mutagenesis to block (C366S) or mimic (C366D) the oxidative modification. While this elicits a binary response in activity, we feel that it is a powerful test to illustrate the direct, causal effect of C366 modification (Figures 6-7). Nevertheless, we recognize that the mutants are not identical to the redox modified Cys, and we have acknowledged this in the discussion (Lines 347-349).

7. In Figure 4F the assessment of NADH activity by DCIP is a poor assay as other enzymes can interfere. This should be repeated with NADH/hexaammineruthenium.

Response: We thank the reviewer for this suggestion. We assessed NADH activity using NADH/hexaammineruthenium as described (PMID: 12351219). We found no effects of light induced ROS generated by *nuo-1::Supernova* on mitochondrial complex I NADH activity (Figure 4F).

8. In Figure 4G, the effect of NAC was unclear as it was present during the light, so the NAC may be blocking the effect, not reversing it. This could be done by blocking with light and then seeing if adding NAC afterwards reversed the effect.

Response: For this experiment, we first illuminated mitochondria and following light treatment we added NAC to reverse the effect (i.e., NAC was not present during the light treatment). We have now expanded our description of the sequence of events in the methods section (Lines 437-441) and the figure legend (Figure 4G, Line 680).

9. The structural model in Fig 6 seemed very speculative to me. While complex I structure is highly conserved I did not feel the benefit of this analysis was worth the speculation – better to wait until there is a C elegans complex I structure, or also consider comparisons with other complex I structures from bacteria and yeasts.

Response: The reviewer is correct, the structural model in Figure 6 is from a recent mouse model of complex I and provides a localization of the mammalian Cys347 in NDUFS2. As it stands, we believe the structural and molecular dynamic data presented in the manuscript adds significant

value, so we are reluctant to remove it entirely. However, we agree it is a bit speculative and can be distracting, so we removed the vibrational modes beyond mode one and the residue cross-correlation matrix presented in (Supplemental Figure 11). To further illustrate the conserved nature of Cys366 in *C. elegans* NDUF-2.1, we include below an expanded multi-sequence alignment comparison of the proximal protein sequence surrounding Cys366 of *C. elegans* NDUF-2.1 to several vertebrates (Supplemental Figure 11A). As expected, the ~40 residues around and including this specific Cys are highly conserved. The recent advance of AlphaFold has enabled us to compare a high confidence structural prediction of the *C. elegans* NDUF-2.1 protein to the solved mouse and human structures (Supplemental Figure 11B). Cys366 / Cys347 is similarly positioned across all three species with a root mean squared distance (RMSD) of under 1 Å. For a relative comparison, the difference between active and deactive complex I in mice is estimated at 2.7 Å (PMID: 33067417). Overall, we believe this along with the included molecular dynamics data provide a reasonable mechanism for how acute ROS can modify the highly conserved Cys366 residue to destabilize Q, impairing overall complex I activity. This data has replaced the previous supplemental figure and we limited the molecular dynamics modeling to the supplement and altered the text (Lines 266-272) to clearly identify the speculation that places our findings in the context of the field.

10. The effects on hypoxia-reperfusion are intriguing. However, to extrapolate this as a mechanism for preconditioning, is a bit of a stretch. There have been many, many models proposed for preconditioning so best not to over interpret these data.

Response: Having worked in the field of ischemic preconditioning for quite some time, we agree completely. Our findings showed that the Cys366 was necessary to mediate responses to hypoxia, and we sought to test if this signaling mechanism would extend other paradigms that involved signaling in response to changes in oxygen, such as ischemic preconditioning. Nevertheless, we acknowledge this point, and have edited the text so as to reduce extrapolation.

Reviewer #2 (Remarks to the Author):

This is an exciting paper, that employs a mix of advanced methods, to propose that a thiol “switch” within complex I is key for hypoxia avoidance behavior in *C. elegans*. Using a CI:SuperNova fusion to enable photo-activatable superoxide formation the authors are able to achieve light-induced avoidance behavior and an increase in speed similar to that induced by hypoxia. By performing state 3/ state 4 measurements on mitochondria isolated from these worms, they observed a decrease in state 3 respiration following CI:SuperNova activation with CI linked substrates but not with CII substrates — leading to the hypothesis that CI linked respiration is somehow halted. The effect of complex I::SuperNova was abolished by treatment with ROS scavengers or loss of *sod-2*, further supporting a role for complex I generated ROS and a reversible modification on CI in mediating the behavioral response. Given that mammalian NDUFS2 has been known to be important in hypoxia signaling, the authors considered three conserved NDUFS2 residues recently reported to be oxidized in a proteomic screen of oxidized cysteines. They focus on Cys366 and create a serine (incapable of getting oxidized) or aspartate (mimicking oxidation) mutant to demonstrate that oxidation of this residue is necessary and sufficient to mediate the behavioral response to SuperNova and hypoxia. They end by showing that oxidation of this residue also mediates ischemic preconditioning. Although some

elements of the core concepts were known in mammalian systems (reversible oxidation of CI cysteines in the context of ischemia reperfusion injury) it is an overall exciting story.

Major critiques:

1. My major concern is related to the purported specificity of Complex I ROS in inducing this behavior. Many perturbations that make worms sick will induce an avoidance response. In fact, the authors do attempt this control, by tethering SuperNova to two distinct places on Complex II. They conclude that the behavioral effect was not recapitulated with complex II:Supernova (lines 132-139). However, in contrast to what is written in the text, the data in the figure (Figure S4) *do* suggest that both of these Complex II constructs have an effect. *sdhc-1::SuperNova* significantly increases body bends in a light-dependent manner, and *sdhb-1::SuperNova* increases body bends at baseline to above that achieved by *nuo-1::SuperNova* activation. If the authors want to claim that ROS generation at Complex I is unique, they need to address this inconsistency.

Response: To further address the specificity of complex I ROS in inducing this behavior we generated two unique light-induced complex III CRISPR lines (*ucr-11::Supernova* and *ucr-2.3::Supernova*). We subjected all of the light-induced electron transport chain ROS lines to both the phototaxis and behavioral assays (Supplemental Figure 5). We define the microdomain effect as having a dose-dependent and phototaxis effect. While there were varying degrees of responses from all the strains, only the *nuo-1::Supernova* had a significantly different phototaxis response and a dose-dependent effect light-dependent response to ROS generation. We have now rephrased to include our definition of selectivity (Results, Lines 143-156) and included these new results in the manuscript (Supplemental Figure 5).

2. I am also concerned about the purported specificity of the C366 residue in mediating the behavioral response to Complex I ROS and hypoxia. Does the C366S mutation block the drop in CI State 3 activity from complex I::SuperNova? Can the authors test other available Complex I mutants to see if they affect the behavioral response to hypoxia or Complex I ROS (as in Figs 7A and 6D, respectively). The *gas-1(fc21)* strain is already used in this study, for example.

Response: Question: “Does the C366S mutation block the drop in CI State 3 activity from complex I::SuperNova?” Figure 6H shows that the C366S mutation blocks the light-dependent decrease in complex activity.

Question: Can the authors test other available Complex I mutants to see if they affect the behavioral response to hypoxia or Complex I ROS (as in Figs 7A and 6D, respectively)? Toxins and mutants lack the ability to dynamically respond to changes in oxygen. Thus, we developed our light-induced complex I ROS model for spatiotemporal control over ROS production. Nonetheless, we subjected two classical complex I and II *C. elegans* mutants, *gas-1(fc21)* and *sdhc-1(kn1)*, respectively, to hypoxia and body bends experiments. As expected, these mutants did not respond to light as in Figure 6D (Supplemental Figure 6). Of note, *gas-1(fc21)* was slow moving and lethargic, as previously documented (PMID:7943840).

Next, we subjected these mutants to acute hypoxia (as in Figure 7A) to test the selectivity of complex I ROS in mediating this effect. The complex II mutant, *sdhc-1(kn1)*, generates ROS from complex II. In agreement with our light-induced complex II ROS system, *sdhc-1(kn1)* responded like wild-type worms to hypoxia (i.e., the hypoxic response is retained). The complex I mutant,

gas-1(fc21) was not able to respond to hypoxia however, as previously documented the mutant had suppressed movement confounding the interpretation.

We have now added this new data (Supplemental Figure 6) and discussed the results (Lines 143-156).

3. The primary phenotype (behavioral response to hypoxia and re-oxygenation) has previously been investigated by the Horvitz lab (PMID: 22405203 and 23811225). These studies identified many genes in the EGL-9/HIF-1 pathway that control this behavior. Can the authors use genetics place their findings in the context of this pathway, e.g., are EGL-9 and CYP-13A12 are required for the behavioral response to Complex I:Supernova? At the very least the authors should discuss in length their work in the context of this important body of work from the Horvitz lab.

Indeed, elegant data from the Horvitz lab demonstrated a role for the well-studied HIF signaling pathway in regulating oxygen dependent behavior. Our results suggest that complex I redox modification may be a proximal sensor for this effect that could act in parallel or upstream of HIF signaling, as a necessary and sufficient signal. As the reviewer requested, we have expanded our discussion to interpret our results in the context of this previous work and included both references (lines 79, 308-317). Additionally, we refer the reviewer to Reviewer 1 comment 1.

4. I found the elastic network modeling very difficult to understand and distracting. This modeling is highly speculative, not needed, and detracts from an otherwise compelling story. I'd urge the authors to consider removing this part of the paper.

Response: We included modeling to give some mechanistic insight on how the modification of a Cys can lead to a change in complex I activity. The model puts our findings in the context of the of a recent publication in the journal (PMID: 33514727). In particular, the model provides a story on how altering a single amino acid results in Q binding site disorder and a decrease in complex I activity. We acknowledge that the modeling is speculative. In response, the modeling is now restricted to the supplement with a simplified, limited discussion (Supplemental Figure 11, Lines 266-272). Additionally, we refer the reviewer to Reviewer 1 comment 9.

Reviewer #3 (Remarks to the Author):

In this manuscript, Onukwufor and colleagues explore the mechanism underlying avoidance behaviors, under hypoxic conditions in *C. elegans*. Triggered by previous reports, which suggest an association between mitochondrial complex I ROS production and hypoxia, the authors tested whether mitochondrial complex I is involved in a specific locomotory response upon hypoxia and reoxygenation. To monitor ROS production in vivo, they created transgenic animals expressing the optogenetic ROS-generating protein SuperNova fused to an endogenous site of ROS production, *nuo-1*, by using the CRISPR-Cas9 system. They found that animals expressing the *nuo-1::SuperNova* sensor showed increased forward movement upon photo-activation as well as increased direction changes compared to wild-type worms. The authors found that the activation of avoidance through *nuo-1::SuperNova* photoactivation was abolished in *sod-2* or/and *sod-3* knockout

strains, suggesting that complex I ROS mediated an increase in locomotion. Conversely, the addition of a SOD mimetic exacerbated light-induced ROS effects and rescued the photo-locomotion response in the absence of SOD. Furthermore, by using a superoxide and catalase mimetic EUK-134 and NAC, the authors obtained similar results since hydrogen peroxide was required for the activation of avoidance upon *nuo-1::SuperNova* activation. Then, the authors focused on the *nduf-2.1* gene (the *C. elegans* orthologue to mammalian NDUFS2 subunit 2), which forms part of the mammalian coenzyme Q-binding site in complex I. In particular, they provide evidence suggesting that Cys366 was both necessary and sufficient to mediate ROS-induced behavioral changes. The Cys366Ser and Asp mutants could not sense acute changes in oxygen; they were also sensitive to hypoxia re-oxygenation (HR) and the degree of protection from hypoxic pre-conditioning (PC) was decreased in these mutant animals. The study suggests that a single thiol residue (Cys366) in the NUDF-2.1 subunit of mitochondrial complex I mediates hypoxic behavioral responses due to reversible oxidation by compartmentalized ROS. As a consequence, complex I enzymatic activity is decreased probably because of destabilization of the coenzyme Q-binding pocket. The manuscript presents some potentially interesting findings and tools. Implementation of a novel optogenetic fusion protein approach to model mitochondria-mediated redox signaling *in vivo* is an added plus. However, key claims made by the authors are not sufficiently supported by data in the manuscript.

1. The authors suggest that their findings (as mentioned above) support a direct role of complex I ROS in mediating hypoxic signaling events. A rigorous support for this notion could come from the use of a *mev-1* mutant strain. *mev-1* encodes the succinate dehydrogenase cytochrome b, a component of complex II, which does not participate in the NADH to oxygen redox/ proton pumping cascade and thus it would not be expected to generate ROS under normal redox conditions.

Response: As suggested, we subjected the classical complex II mutant, *mev-1(kn1)* (a.k.a., *sdhc-1(kn1)*) to acute hypoxia (see response to Reviewer 2, comment 2). *mev-1(kn1)* generates complex II ROS at baseline and we found no changes to the hypoxia response (i.e. *mev-1(kn1)* responded to hypoxia as wildtype). These results (Supplement figure 6) support the notion that complex I ROS is selective in mediating the behavioral change.

We further demonstrated the selectivity of complex I ROS in mediating this effect by generating additional light-induced electron transport chain models (see response to Reviewer 2, comment 1). These models showed that complex I ROS, not complex II nor complex III, selectively elicited the behavioral phenotype.

2. Also, with respect to the regulation of hypoxic avoidance behavior by complex I ROS, the authors suggest that this idea is further supported by the fact that complex II::SuperNova photo-activation failed to produce a behavioral response similar to that produced by complex I::SuperNova photo-activation. To strengthen their statement, they should test the behavior of animals expressing a complex III::SuperNova transgene since complex I and III are the main sites of ROS production in mitochondria.

Response: We thank the reviewer for this suggestion and generated two additional CRISPR lines to control complex III ROS with light (see Reviewer 2 comment 1). Both *ucr-11::SuperNova* and *ucr-2.3::SuperNova* strains had no effect on locomotion or phototaxis (Supplement Figure 5) in

response to light. Collectively, these results support the specificity of complex I ROS in mediating the behavioral change.

3. In Figure 1A, a superoxide dismutase and catalase mimetic EUK-134 renders nematodes unresponsive to changes in oxygen levels, presumably by sequestering ROS. However, the reverse experiments using paraquat and rotenone to elevate mitochondrial ROS production have been conducted under baseline conditions (figure 1B-C). The authors should specifically address whether higher ROS are accompanied by increased motility when animals face a hypoxia-reoxygenation stress and perform a longitudinal locomotion analysis, similarly to figure 1A.

Response: The reviewer suggests testing if toxin-induced ROS from paraquat is additive to hypoxia-induced ROS in mediating the behavioral response. We subjected paraquat treated worms to acute hypoxia. We found that paraquat treated worms, while initially fast, had a suppressed behavioral response to hypoxia. Upon re-oxygenation, the worms gradually returned to the faster locomotion seen pre-hypoxia (Supplement Figure 1E). These results highlight the difference in acute vs sustained ROS production. We hypothesize that the higher baseline ROS levels could prevent the dynamic changes necessary to respond to hypoxia. Alternatively, the higher levels of ROS could elicit compensatory mechanisms that block the acute response to hypoxia-mediated ROS. This paraquat response supports the genetic phenotype of the C366D, whereby the constitutively active mutant is not able to respond to changes oxygen concentration (Figure 7A).

Overall, these results support the development of a model with spatiotemporal control of ROS seen with *nuo-1::Supernova*. We included this new data (Supplement Figure 1E) and expanded the discussion (Lines 93-101, 285-287).

4. In figure 2, following an HPLC-based approach, the authors validate that *nuo-1::Supernova* transgenic can generate excessive ROS upon illumination with a light source. They can use additional methods to corroborate this finding. For example, they can measure ROS in worm lysates using CM-H2DCFDA (PMID: 29707606) or utilize fluorescent dyes (such as Mitotracker ROS) to quantify mtROS *in vivo*. Better quality and higher magnification images would be more informative.

Response: As described in PMID: 22027063, DCFDA is not a selective measure of superoxide or hydrogen peroxide and is artifact prone. We believe the reviewer is referring to mitoSOX as a fluorescent measure of mitochondrial superoxide *in vivo*. mitoSOX is a mitochondria-targeted version of dihydroethidium, which is the compound used in our studies. However, fluorescence alone is not sufficient to detect superoxide since the superoxide selective oxidation product (2-OHE+) is spectrally indistinguishable from the nonselective oxidation product, ethidium. HPLC separation of the dihydroethidium oxidation products is required to rigorously measure superoxide. We refer the reviewer to many reviews on this topic (e.g. PMID: 22027063).

In our studies, we used state-of-the-art and the current gold standards in the redox field. We measured mitochondrial hydrogen peroxide *in vivo* using HyPer7 and superoxide using HPLC separation of the superoxide-selective dihydroethidium oxidation product 2-OHE+. Importantly, we do not claim excessive ROS associated with oxidative stress are mediating the behavioral effect. Rather, we claim localized low levels of ROS generation.

5. In Figure 2B-C, the authors are using Paraquat and rotenone in order to investigate whether ROS originating from complex I cause changes in locomotion behavior. They should show whether these drugs lead to the same or additive effects regarding locomotion under hypoxic conditions.

Response: We assume the reviewer is referring to Figure 1B-C using paraquat and rotenone and investigating body bends under hypoxic condition. We agree and have carried the experiment as suggested by reviewer (see Reviewer 3, comment 3).

6. In Supplementary Fig. 4C, the authors investigate whether light exposure of complexII::supernova induces similar effects to those of complex I::supernova. At least for the sdhc-1::supernova, it seems that mild light exposure increases body bends. Therefore, it could be that ROS generation from complex II may also contribute to the behavioral change. The fact that there is not dose-dependent effect in higher concentrations is adequate to exclude complex II contribution? Do the authors have any indication for the involvement of other ETC complexes except for complex I and II?

Response: We have generated additional CRISPR lines to test other electron transport chain sites of ROS production and subjected all the strains to phototaxis experiments (see reviewer 2, comment 1, and Reviewer 3, comment 2). In summary, the results support the specificity of complex I ROS in mediating the behavioral change.

7. ROS levels have been associated with lifespan and C. elegans mutants for nduf-2.1 (a.k.a gas-1) are short lived. The authors have generated a redox-insensitive C366S mutant. It would be interesting to compare the lifespan of those mutant animals, to that of the sulfenic acid modification mimetic (C366D) mutants, which phenocopies nduf-2.1 depletion. Could the identity of the C366 residue determine lifespan?

Response: We thank the reviewer for this suggestion. This is a new direction that we will pursue with future studies. However, we believe the role of complex I redox on lifespan is beyond the scope of this manuscript.

8. The authors have not at all discussed the widely accepted phenomenon of mitohormesis, which actually represents the positive signaling role of mild ROS production from mitochondria upon caloric restriction. Specifically, it has been shown that caloric restriction induces a transient ROS signal that improves stress resistance leading to increased mitochondrial bioenergetics (PMID 17908557, 21619928), contrary to what is shown here (ROS production decrease mitochondrial bioenergetics).

Response: The difference between mitohormesis, as seen in models of caloric restriction, and our current findings is the result of acute vs chronic ROS production. Much like other second messengers, such as calcium, the timing duration and amount of ROS can influence the physiologic outcome PMID: 24563855. Long-term mild ROS production can elicit transcriptional changes resulting in the increased expression of stress resistance proteins. This process requires protein synthesis and requires hours to days to develop. Acute ROS signaling, as seen in our study, is the result of local, small changes in ROS. The signaling and response occurs on the

scale of seconds to minutes and is rapidly reversed. It is important to note that the observed decrease in bioenergetics is reversible.

We have now included a discussion on the diverse role ROS play in physiology (Lines 308-317).

9. The authors show that light affects ROS production even in the absence of SuperNova (Figure S3D). However, light does not induce an increase in body bends in wt worms (Figure S4C)? On the other hand, illumination almost doubles survival, even when imposed hours before hypoxic treatment. These findings question the relevance of the body bend assay used in this study.

Response: Figure S3D (Now Supplemental Figure 4) uses dihydroethidium, a fluorescent molecule to measure ROS. Dihydroethidium is light sensitive (PMID: 31841676) and light causes an increase in DHE oxidation products in wild-type worms. Our data demonstrate the body bend response requires ROS generated in the complex I microdomain (Figure 2). The light affects on DHE generates a ROS signal independent of *nuo-1::Supernova* and is not generated at complex I, hence as expected there is no increase in body bends in wild-type worms.

ROS detection methods are not without caveats and we have expanded our discussion on these experiments (Lines 371-372).

10. The authors show that it is hydrogen peroxide generated from superoxide anion from the Supernova reporter fused to NUO-1 that affects the redox state of the cysteine366 of NDUFS2, as SOD-2 or SOD-3 depletion reverses the effects of light. However, if it is not the spatial proximity of generated superoxide anions that is responsible for Cys366 oxidation, then one would expect that superoxide anions generated in the matrix from the SN fusion to complex II would also have the same effects (increased hydrogen peroxide production and Cys366 oxidation). However, this is not the case, as SN fused to complex II failed to induce similar phenotypic effects. Why is this?

Response: Our results demonstrate that the complex I ROS microdomain mediates a behavioral response. This is due to a privileged ROS microdomain whereby low levels of complex I ROS act locally to modify a complex I subunit. Our data suggests that the generation of superoxide and conversion to hydrogen peroxide in the proximity complex I is required. In further support of a privileged microdomain, SOD-2 is localized to complex I supercomplexes (PMID: 23895727).

We acknowledge that the overproduction of ROS at any site eventually modify Cys366 as well as indiscriminately modify other proteins, DNA and lipids. However, this will not result in a selective response but would rather lead to oxidative damage and death. We have now included this in our discussion (Lines 338-341)

11. It is known that C347 oxidation status affects Complex I protein stability in mammals (PMID 27052170). Does SN activation affect protein levels of Complex I? Could this be the reason for reduced Complex I oxygen consumption? What are the effects of Complex II-fused SN on complex I stability? Western blot analysis is required.

Response: It is important to note, PMID 27052170 demonstrated that oxidation of complex I can proceed degradation of complex I in response to hypoxia-reoxygenation. Since hypoxia-

reoxygenation can generate large amounts of ROS, this study found numerous Cys residues on many complex I subunits that are modified under these conditions. The study did not show that NDUFS2 C347 was necessary or sufficient for complex I degradation and the role for C347 is unclear.

Q: “Does SN activation affect protein levels of Complex I?” “Could this be the reason for reduced Complex I oxygen consumption?” We found reduced complex I oxygen consumption and activity in response to *nuo-1::Supernova* activation. This loss of activity is independent of complex I degradation for the following reasons.

- 1) The complex I enzyme activity assay is from isolated complex I and is independent of the proteolytic machinery (Figure 4).
- 2) The NADH activity of isolated complex I is not effected by *nuo-1::Supernova* activation (Figure 4F and Supplemental Figure 8). This result would not be possible if the entire complex was degraded.
- 3) The loss of total complex I activity in response to *nuo-1::Supernova* activation was recovered by the addition of NAC (Figure 4G).
- 4) The behavioral changes were plastic (i.e., reversible) and returned to baseline over the course of seconds (Figure 4H). This time course is not compatible with the synthetic nature of protein synthesis required to remake complex I (NDUFS2 protein turnover is on the scale of days, PMID: 22915825).

In addition to the above lines of evidence, we also directly tested if *nuo-1::Supernova* activation affected complex I protein levels using western blot. We illuminated isolated mitochondria and used high resolution clear native electrophoresis (hrCNE, PMID: 17426019) to measure the complex I levels. We found the light-induced ROS did not change the levels of complex I (Supplemental Figure 9).

Q: “What are the effects of Complex II-fused SN on complex I stability?” Complex II ROS did not have an effect on the behavioral response and thus this site-specific ROS production was not pursued further in this context.

12. The authors state that they identify a reversible mitochondrial complex I thiol switch. However reversibility is poorly shown. They only show that locomotion returns to normal just within a few seconds without light (Figure 3H). Based on this finding the authors suggest that thiol oxidation is reversed and ROS/H₂O₂ production as well but this is not shown. The conclusion that: “Collectively, these results suggest a reversible redox modification in complex I downstream of the NADH-binding site” is an overstatement not supported by the data shown here.

Response: We have now strengthened our evidence to support the reversibility of the effect and have the follow data.

- 1) Light-induced behavioral response is reversed when light is removed and the response rapidly returns to normal within 30 seconds (Figure 3H).
- 2) The light-induced response can be re-activated after an animal returned to baseline (Figure 3H).
- 3) Light-induced complex I inhibition is reversed following the addition of a thiol antioxidant (Figure 4G).
- 4) Light-induced complex I inhibition is not mediated through degradation (Supplemental Figure 9).

5) Light-induced responses require ROS formation and are blocked by chemical and genetic perturbations (Figure 5 and Supplemental Figure 10).

6) CRISPR/Cas9 mutation show that redox modification of *nduf-2.1* Cys366 is both necessary and sufficient to induce behavioral changes (Figures 6 and 7). *nduf-2.1* is downstream of the NADH-binding site and forms part of the Q-binding pocket.

These separate lines of evidence support the conclusion that a reversible redox modification in complex I downstream of the NADH-binding site mediates the behavioral response.

Minor comments

13. It would interesting to check whether complex II ROS production from SDHC-1::Supernova and SDHB-1::Supernova will exhibit increased phototaxis avoidance behaviors similar to NUO-1::Supernova upon light titration.

We have now included these results (Supplemental Figure 5D-E).

14. How does this redox modification to a single cysteine residue in NDUF2.1 affect other types of stress responses?

We found that NDUF-2.1 Cys366 is required for the response to rotenone (Figure 6E).

15. For the mitochondrial bioenergetic studies, it is not clear whether illumination persists throughout the oxygen consumption assay or whether it precedes it. In general, the authors need to provide a more detailed description of the protocols and procedures used.

Illumination occurred prior to oxygen consumption measurements. We have now added more details to the procedure (Lines 173-174,428-429).

16. The authors have to describe in a more concise and comprehensive way which transgenic animals they are using. For instance, in the legend of Figure 2D they refer to *sdhc-1::mCherry*, while in the results section of the main text, the particular transgene is not mentioned at all.

We have now expanded the description of the constructs in a concise manner. In Figure 2D, *sdhc-1::mCherry* localization was previously characterized PMID: 30887829 and it was included as a control.

17. In figure 2B, the magnified images are blurry and should be replaced with images of higher quality.

We have now included higher resolution images.

18. In the graph of figure 2E, “p=0.12” should be changed to “ns”.

We have made the suggested change.

19. Supplementary Figures should be also mentioned in the main text. For instance, Supplementary Fig. 1 is only mentioned in the figure legend of Figure 1. Several figures do not appear in text according to their numerical order.

We now mention all supplementary figures in the main text.

20. Sup. Fig. 1: Paraquat description is missing in the legend, and panel order is scrambled.

We have now corrected the figure legend.

21. The graphs A-D in supplementary Figure 3 should be properly aligned.

Supplementary Figure 3 (Now Supplementary Figure 4) is aligned.

22. Figure 4: no annotation of time on x-axis

The x-axis is now corrected.

23. Figure S7C is not clear

We have now simplified Supplemental Figure 7 (now Supplemental Figure 11).

24. Proper nomenclature for gene names, mutant strains, etc., should be observed throughout.

We adhere *C. elegans* and mammalian nomenclature PMID: 29722207 and have corrected throughout the text in Table 1.

REVIEWERS' COMMENTS

Reviewer #1 (Remarks to the Author):

I felt that the authors made a good job of addressing my criticisms. The rationale for the incorporation of the supernova -NUo1 was presumably that the wt nuo-1 in knocked out so the worms are only viable if the construct is incorporated into complex I. Apologies that I missed that first time, but even so the native gel exps do add to the paper.

Reviewer #3 (Remarks to the Author):

The revised version of the manuscript is significantly improved. The authors have made a commendable effort and have adequately answered to all the comments. They managed to obtain two additional CRISPR/Cas9 lines in order to exclude possible involvement of other ETC complexes in the regulation of hypoxic avoidance behavior. Results are adequately discussed and further clarifications are provided. Overall, this study provides novel mechanistic insights regarding the nematode avoidance behavior upon hypoxic conditions. The method for spatiotemporal control of ROS generation is versatile and could be easily applied in the future to delineate the effects of ROS at specific intracellular sites of interest.

Some points were not exhaustively examined as for example, the point 14 of Reviewer #3. In this case, the authors found that NDUF-2.1 Cys366 is required for the response to rotenone, besides the behavioral response to hypoxia. Additional stresses, which are known to increase ROS generation, such as high glucose or fatty acids, could have also been examined. Nevertheless, this is a minor comment.

Moreover, the authors did not test whether the single thiol residue in mitochondrial complex I (C366) can also influence lifespan. However, this could be an objective of a future study, as they stated.

Figure 3H. Second graph at 75 s: The authors state (at line 635) that “for the reversal assay, the light source was removed and body bends were scored every 15s for 60s. For plasticity, the light was removed for 60s, then reintroduced and body bends were scored for 15s.” Are both assays in the second graph? Are they using different worms?

Line 220: the word “dismutase” is missing, so please rephrase the sentence as “using a superoxide dismutase/catalase mimetic EUK-134”

We thank the editor and reviewers for their comments and agree that the manuscript is stronger as a result. We have reproduced the reviewer comments below in bold and provided responses in regular typeface.

Reviewer #1 (Remarks to the Author):

I felt that the authors made a good job of addressing my criticisms. The rationale for the incorporation of the supernova -NUO1 was presumably that the wt nuo-1 in knocked out so the worms are only viable if the construct is incorporated into complex I. Apologies that I missed that first time, but even so the native gel exps do add to the paper.

We thank the reviewer for the positive comments and agree that the native gels have made the manuscript stronger.

Reviewer #3 (Remarks to the Author):

The revised version of the manuscript is significantly improved. The authors have made a commendable effort and have adequately answered to all the comments. They managed to obtain two additional CRISPR/Cas9 lines in order to exclude possible involvement of other ETC complexes in the regulation of hypoxic avoidance behavior. Results are adequately discussed and further clarifications are provided. Overall, this study provides novel mechanistic insights regarding the nematode avoidance behavior upon hypoxic conditions. The method for spatiotemporal control of ROS generation is versatile and could be easily applied in the future to delineate the effects of ROS at specific intracellular sites of interest.

Some points were not exhaustively examined as for example, the point 14 of Reviewer #3. In this case, the authors found that NDUF-2.1 Cys366 is required for the response to rotenone, besides the behavioral response to hypoxia. Additional stresses, which are known to increase ROS generation, such as high glucose or fatty acids, could have also been examined. Nevertheless, this is a minor comment.

Our study tested the hypothesis that complex I ROS generation plays a role in hypoxic signaling. Rotenone is known to increase ROS at complex I. The mechanisms of ROS production during high glucose or fatty acid oxidation are distinct and beyond the scope of a hypoxic signaling study. However, it should be noted that in response to the reviewer's previous comments, we tested the role of other sites of mitochondrial ROS production *directly* using newly generated CRISPR alleles to control ROS production using light. These new tools allowed us to *directly* control ROS production at other sites using light as opposed to *indirectly* generating ROS by changing in vivo glucose or fatty acids levels.

Moreover, the authors did not test whether the single thiol residue in mitochondrial complex I (C366) can also influence lifespan. However, this could be an objective of a future study, as they stated.

We agree with the reviewer that aging experiments are beyond the scope of our hypoxic signaling study and would be a completely distinct line of research.

Figure 3H. Second graph at 75 s: The authors state (at line 635) that “for the reversal assay, the light source was removed and body bends were scored every 15s for 60s. For plasticity, the light was

removed for 60s, then reintroduced and body bends were scored for 15s.” Are both assays in the second graph? Are they using different worms?

The reviewer is correct and both assays are in the second graph using different worms. We have now clarified the text to better describe the groups and figure.

Line 220: the word “dismutase” is missing, so please rephrase the sentence as “using a superoxide dismutase/catalase mimetic EUK-134”

We have now corrected the text.